# Neural coding of choice and outcome are modulated by uncertainty in orbitofrontal but not secondary motor cortex

Juan Luis Romero-Sosa[1] ✉, Alex Yeghikian[1], Andrew M. Wikenheiser[1,2,3], Hugh T. Blair[1,2,3,5] & Alicia Izquierdo [1,2,3,4,5] ✉

Orbitofrontal cortex (OFC) and secondary motor cortex (M2) are both implicated in flexible reward learning but the conditions that differentially recruit these regions are not fully understood. We imaged calcium activity from single neurons in rat OFC or M2 during de novo learning of increasingly uncertain reward probability schedules. Predictions of choice were decoded from M2 neurons with high accuracy under all certainty conditions, but were more accurately decoded from OFC neurons under greater uncertainty. Decoding accuracy of choice and outcome was predicted by behavioral strategies Win-Stay and Lose-Shift in OFC, but not M2. Whereas chemogenetic inhibition of OFC neurons attenuated learning across all schedules, M2 neurons were found to support learning in only the most certain reward schedule. Thus, OFC neurons preferentially encode choices and outcomes that foster a greater reliance on adaptive strategies under uncertainty. This reveals a functional heterogeneity within frontal cortex in support of flexible learning.

Both the orbitofrontal cortex (OFC) and secondary motor cortex (M2) are involved in flexible reward learning and reward-motivated decision making. M2 receives dense inputs from diverse cortices, including sensorimotor, visual, retrosplenial, and orbital regions[1], and in turn, sends projections to the striatum that fall along mediolateral and rostrocaudal gradients within M2[2], enabling precise control of actions. This anatomical connectivity positions M2 well to control action flexibly based on both sensory and motivational information. Functional studies have determined that M2 is essential for action updates[3] and that it enables the flexible assignment of either stimuli or motor antecedents to action plans[4–6]. Additionally, there has been a discovery of different population dynamics in M2 depending on the strategy and stimulus modality that mice use in a flexible decision-making task[7]. Given its integration of sensory and motivational evidence, M2 has an essential role in higher-order movement control and orienting[8], but can also bias choice based on outcomes and choice history[9,10]. Though a previous electrophysiological study indicated OFC is involved in

updating outcome value from prior actions[11], authors found that M2 signaled choice earlier than any other area of rat frontal cortex, and remarkably also contained a value signal[12].

OFC also has a well-substantiated role in flexible decision making[13–16]. Several groups have reported choice-predictive neurons in OFC in equivalent or even larger numbers than value or outcome-encoding cells in this region[17–21]. One group found that over half of the neurons they recorded in OFC encoded both the direction of choice and the location of the goal, concluding that choice-selective neurons in OFC represent spatial-motor information required to define specific goals and necessary actions to obtain them[21]. However, it is presently unclear if and how these representations change with uncertainty. By uncertainty, we mean stochasticity, or moment-to-moment probabilistic outcomes[22] generating an "expected uncertainty"[23–25], not uncertainty resulting from volatility, or the rate of change[22]. Indeed, a somewhat parallel literature suggests OFC may be critically involved in learning, especially under uncertain or probabilistic conditions[26–28],

[1]Department of Psychology, University of California-Los Angeles, Los Angeles, California, USA. [2]The Brain Research Institute, University of California-Los Angeles, Los Angeles, California, USA. [3]Integrative Center for Learning and Memory, University of California-Los Angeles, Los Angeles, California, USA. [4]Integrative Center for Addictions, University of California-Los Angeles, Los Angeles, California, USA. [5]These authors contributed equally: Hugh T. Blair, Alicia Izquierdo. ✉e-mail: jromero12@ucla.edu; aizquie@ucla.edu

where it may set the expectation for the rest of learning[24,25,29,30]. This is consistent with the idea that OFC stores long-term action values[24,31], stimulus values[32], and causally contributes to an expected uncertainty or baseline risk in rats[33,34]. This risk is signaled by single neurons in OFC in rats[35,36] and in nonhuman primates[37,38]. One study[39] reported that OFC neurons track expectations across a slow timescale (i.e., across-session probabilistic reversal learning) and are also recruited for trial-by-trial learning, or value updates, on a fast timescale (i.e., within a session). However, this study did not manipulate the reward uncertainty over time. Critically, aside from the Hattori et al. (2023) study, recording and imaging studies in both rodent and primate species typically assess the role of OFC in decisions in expert animals, not during the learning of risk or expected uncertainty. Thus, it is unknown how choice and outcome cell selectivity may change over the course of learning, especially with increasing uncertainty in OFC. All of the above suggest that although M2 and OFC are involved in flexible learning and decision-making, their involvement could dissociate over different timescales and increasingly uncertain conditions.

Using miniscopes, we compared populations of neurons and single cells in OFC and M2 over several sessions of learning in freely behaving rats. Unlike the great number of studies assessing these regions in well-trained animals, here we studied calcium dynamics during de novo learning. We were primarily interested in determining if and when neural dynamics would differentially predict choice and outcome in M2 vs. OFC, and which behavioral and task features single cells were selective for in these regions across learning under uncertainty.

## Results

### Learning under increasing uncertainty

Male and female Long–Evans rats first underwent stereotaxic surgery to infuse GCaMP6f into either M2 or OFC (see "Methods"). In the same surgery, a GRIN lens was unilaterally implanted over the site of the viral infusion (Fig. 1I–K; photomicrographs of lens implant and calcium indicator expression for all rats are shown in Supplementary Fig. 1). Following surgical recovery and baseplating, rats then underwent pre-training followed by six sessions of learning. In a touchscreen-outfitted chamber, rats were required to initiate each trial by touching a center stimulus and then make a choice to touch a stimulus on either the left or right side. If correct, rats could retrieve a sucrose pellet reward from a food port located on the opposite side of the touchscreen response zone, signaled by a light cue in the port. This visual cue appeared concurrently with pellet delivery on rewarded, but not unrewarded, trials. The reward contingencies reversed every 75 trials and increased in uncertainty on the last block of trials, with reward schedules progressing from less to more reward uncertainty (100:0 to 70:30) (Fig. 1A, B). The probability of reward delivery depended on the rat's choice (left vs. right), with one side delivering a higher probability of reward than the other. Each session consisted of 3 blocks of 75 trials (225 trials total); the response with the highest reward probability (left vs. right) was reversed at the start of each block. Within each session, the probability ratio remained the same for the first two trial blocks and then changed to a ratio with higher uncertainty for the last block of trials. As shown in Fig. 1C, sessions 1 and 2 used a ratio of 100:0 for blocks 1-2 and 90:10 for block 3 (Schedule 1), sessions 3 and 4 used a ratio of 90:10 for blocks 1-2 and 80:20 for block 3 (Schedule 2), and sessions 5 and 6 used a ratio of 80:20 for blocks 1-2 and 70:30 for block 3 (Schedule 3). Figure 1D shows an example of a single rat's performance during a Schedule 2 session.

Results of an omnibus Generalized Linear Model (GLM) with a binomial distribution on predictors of raw choices (0, 1) are found in Supplementary Table 1. The main dependent variable was whether the rats chose the higher reward probability location. Rats improved on this measure across trials (GLM: $\beta_{trialnum} = 0.0559$, $p < 0.001$) and blocks ($\beta_{block} = 1.6671$, $p < 0.001$), but not across schedules ($\beta_{schedule} = 0.1454$, $p = 0.58$), Fig. 1E. The M2 lens-implanted group

exhibited attenuated learning compared to the OFC lens-implanted group ($\beta_{area} = 1.6671$, $p = 0.0416$) (Supplementary Table 1).

Implanted animals also exhibited slightly attenuated learning compared to viral, non-tethered controls (GLM: $\beta_{Implanted\_Lens} = -0.18$, $p < 0.01$, Supplementary Fig. 2 and Supplementary Table 28). For latency measures, non-tethered control animals initiated trials and collected rewards more quickly but exhibited slower choice latencies (Supplementary Tables 29–31). Latencies to initiate trials and collect reward did not differ in M2- vs. OFC-implanted animals, but choice latencies were faster in the OFC-implanted animals than in the M2-implanted animals (GLM: $\beta_{Area} = -0.57$, $p < 0.01$, Fig. 1F–H and Supplementary Tables 2–4).

### Decoding choice from calcium activity

Binary support vector machine (SVM) classifiers were trained to predict *Chosen Side* (left vs. right) from calcium traces aligned to the choice nosepoke (Fig. 2A, B). To determine the most appropriate windows for statistical analysis, we generated peak-aligned heatmaps and found cells that exhibited time-locked activity to either choice or reward cue (Fig. 2D, E). Decoder training sets were balanced to include an equal number of rewarded/unrewarded and left/right trials across all schedules (Fig. 2C). In both M2 and OFC, choice decoding accuracy ramped up from chance starting prior to trial initiation (suggesting that rats often anticipated which side they would choose prior to trial initiation), and peaked ~ 500 ms after choice (Fig. 2F). An analysis of predictors of decoding accuracy in the bin just prior to choice (yellow shaded region in Fig. 2F) resulted in more accurate decoding in M2 than OFC ($\beta_{Area} = -0.231$, $p = 0.0004$), with no effect of schedule ($\beta_{Schedule} = -0.0843$, $p = 0.19$), but a significant Area x Schedule interaction ($\beta_{Area:Schedule} = 0.056$, $p = 0.0475$), Supplementary Table 5. *Chosen Side* was decoded from M2 with similarly high accuracy under all probability schedules, whereas decoding was more accurate in OFC when the schedule was more uncertain. Confirming this, post-hoc analyses on each schedule (Supplementary Table 6) revealed that decoding from M2 was more accurate than from OFC under conditions of higher certainty (Schedule 1: $\beta_{Area} = -0.174$, $p = 0.0027$; Schedule 2: $\beta_{Area} = -0.119$, $p = 0.0438$), but under the least certain schedule, OFC decoding accuracy increased to become equivalent to M2 decoding accuracy (Schedule 3: $\beta_{Area} = -0.0621$, $p = 0.073$). This certainty-dependent decoding accuracy in OFC was not accounted for by experience (i.e., session) since experience was a covariate in the GLM model and was not a significant predictor of choice accuracy ($\beta_{Session} = 0.0268$, $p = 0.234$). In addition, to check if the number of neurons used to train decoders affected decoder accuracy, we entered this variable as a moderator in the GLM. We did not find any significant effect or interaction with any other variable ($\beta_{NumCells} = -0.0008$, $p = 0.22$, $\beta_{NumCells:Area} = 0.0008$, $p = 0.12$, $\beta_{NumCells:schedule} = 0.0004$, $p = 0.17$, $\beta_{NumCells:Area:schedule} = -0.0003$, $p = 0.19$), indicating that ensemble size did not significantly explain decoder accuracy results.

### Decoding outcome from calcium activity

SVM classifiers were trained to predict *Trial Outcome* (rewarded vs. unrewarded) from calcium traces aligned to the choice nosepoke (Fig. 2A, B). At time points before the outcome was known (that is, prior to 1 s after the choice nosepoke), the accuracy of outcome decoding was limited by the certainty of the outcome (which was highest under Schedule 1 and lowest under Schedule 3). In both M2 and OFC, decoding accuracy for *Trial Outcome* began ramping up from chance at trial initiation and then stalled at a plateau, which persisted until the outcome was known, after which decoding accuracy began ramping upward again (Fig. 2G). A GLM analysis of decoding accuracy in the bin just prior to the outcome (yellow shaded region in Fig. 2G) resulted in, as expected, better decoding of outcome for certain than uncertain schedules ($\beta_{Schedule} = -0.128$, $p = 0.0494$). Decoding was also more accurate in M2 than OFC ($\beta_{Area} = -0.138$, $p = 0.0248$). There

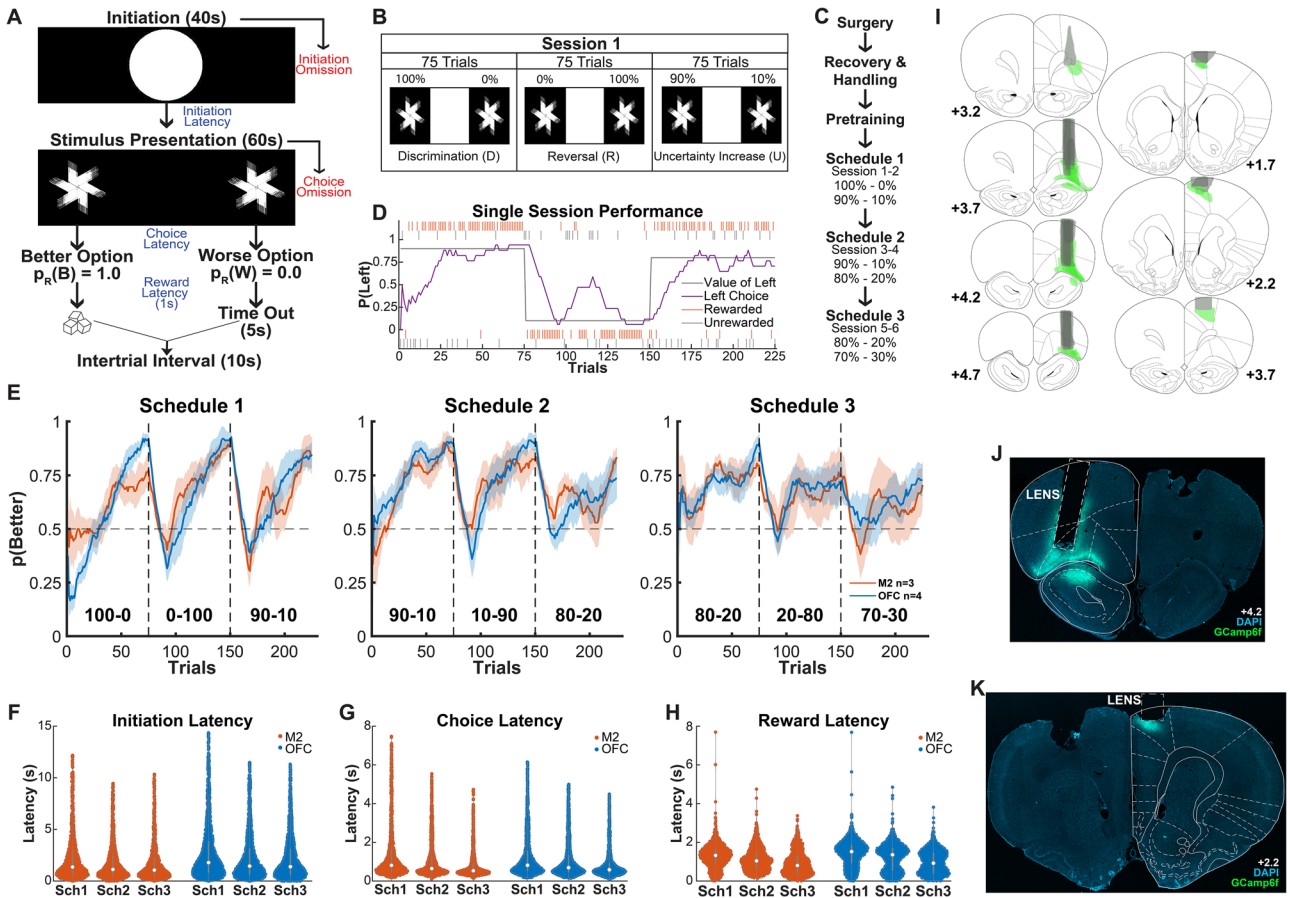

**Fig. 1 | Probabilistic reversal learning task performance in freely-behaving, lens-implanted rats. A** Schematic of trial timeline: Rats learn in an operant chamber outfitted with a touchscreen with three different interaction zones (left, center, and right) and a reward port on the opposite side of the chamber. A white circle is presented in the middle zone for 40 s, during which the rat is required to nosepoke to initiate a trial. Subsequently, two identical stimuli are presented on the left and right sides of the screen for 60 s. The rat is required to nosepoke one of these stimuli, selecting either the left or right side. One of the sides has a higher reward probability than the other. The rat either gets rewarded 1 s after the nose-poke or undergoes a 5-second timeout. There is a 10 s Intertrial Interval (ITI) before a new trial begins. **B** A session consists of 3 blocks of 75 trials each, with two reversals, with the last block increasing in uncertainty. **C** Following surgery, recovery, handling, and pretraining to respond to touchscreen stimuli on either the left or the right, different schedules are introduced, with each schedule administered across two days each, and increasing in uncertainty. **D** Sample performance on Schedule 2, where the rat matches the value of the left side and adapts its behavior following a reversal. **E** Probabilities of choosing the better option, mean ± SEM (shading). There was a significant difference in accuracy, p(Better), by lens implant area (Supplementary Table 1). **F–H** Performance measures by lens implant region to include latencies for Initiation (**F**) group sizes are: $n = 1286$, $n = 1286$, $n = 1301$, $n = 1716$, $n = 1717$, $n = 1707$ respectively; Choice (**G**) group sizes are: $n = 1284$, $n = 1291$, $n = 1305$, $n = 1727$, $n = 1730$, $n = 1736$ respectively; and Reward (**H**) group sizes are: $n = 830$, $n = 851$, $n = 668$, $n = 1149$, $n = 1172$, $n = 1023$ respectively for M2- and OFC-lens implanted rats across all the schedules. The center of each box plot is the median. **I** Reconstructions of calcium indicator GCaMP6f and GRIN lens implant in all rats. **J** Photomicrograph of a lens in OFC. **K** Photomicrograph of a lens in M2. Coronal sections reprinted from The rat brain in stereotaxic coordinates, 7th edition, Paxinos, G. & Watson, C., 2014, with permission from Elsevier. Source data are provided as a **Source Data file**.

was similarly no effect of overall experience, or session number (Supplementary Table 7). Here, we also included the number of neurons as a moderator and again did not find a significant effect or interaction of ensemble size with any of the other predictors ($\beta_{NumCells} = -0.00001$, $p = 0.98$, $\beta_{NumCells:Area} = 0.0002$, $p = 0.59$, $\beta_{NumCells:schedule} = 0.0002$, $p = 0.34$, $\beta_{NumCells:Area:schedule} = -0.0002$, $p = 0.31$). The same analysis was conducted on only co-registered cells (i.e., the decoder was trained on session 1 and tested on session 2, within the same schedule), Supplementary Fig. 6. This revealed a similar pattern as Fig. 2F and pointed to greater stability for Chosen Side decoding over Trial Outcome.

### The ratio of choice, outcome, and reward-selective neurons increases with uncertainty in OFC, but not in M2

To link the decoder results to single-cell selectivity, we removed an increasing number of cells with large beta coefficients (Supplementary Fig. 3). As expected, decoder accuracy steadily dropped as more

neurons with the largest beta coefficients were removed. We show heatmap plots of average trial activity for neurons that were significantly selective ($p < 0.05$) for *Chosen Side*, *Trial Outcome*, and *Reward Retrieval* (Fig. 3A–C), as well as trial-by-trial raster plots of calcium activity from individual example cells (Fig. 3G–I).

M2 contained a higher ratio of neurons selective for *Chosen Side* than OFC throughout all schedules (GLM, $\beta_{Area} = -0.238$, $p < 0.01$). There was a significant interaction between area and schedule (GLM, $\beta_{AreaxSchedule} = 0.063$, $p = 0.007$) (Supplementary Table 8), reflecting the fact that the ratio of choice-selective neurons increased in OFC with uncertainty (GLM, $\beta_{Schedule} = 0.06$, $p < 0.01$, Fig. 3D), but was not influenced by uncertainty in M2 (GLM, $\beta_{Schedule} = 0.004$, $p = 0.78$) (Supplementary Table 9). Session (i.e., experience) was not a significant predictor of cell selectivity. Thus, the ratio of choice-selective neurons increases in OFC with uncertainty, but does not change with uncertainty in M2.

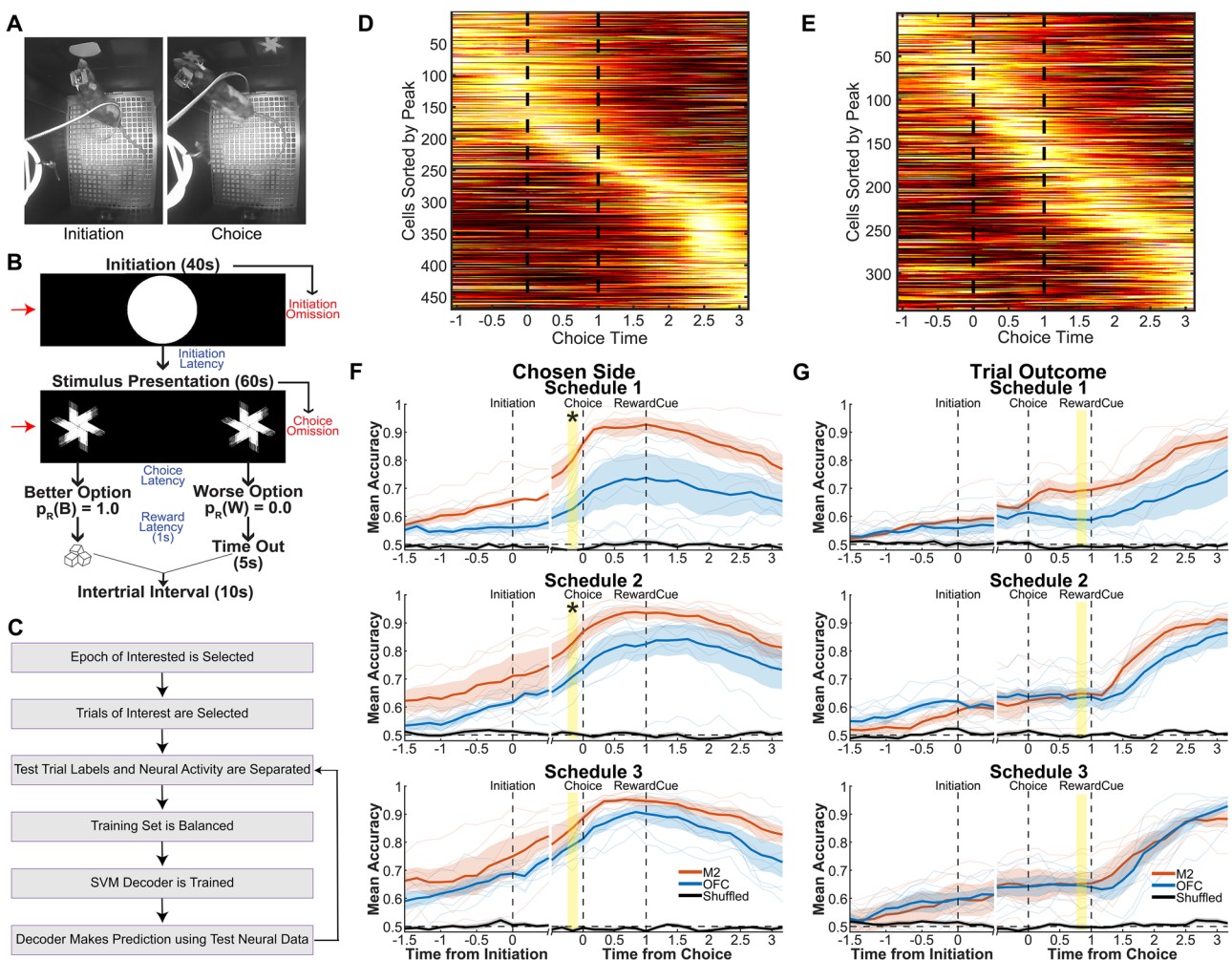

**Fig. 2 | OFC, but not M2, choice decoding accuracy increases with uncertainty.**
**A** Trial epochs selected for analysis of neural data: Initiation (left) and Choice
(right). **B** Same epochs as A, shown in the trial timeline. **C** A flowchart of how the
decoder was trained with balanced data and then tested on each trial. **D** Example of
a cross-validated peak-aligned heatmap aligned at choice from an imaging session
in M2 during schedule 2. **E** Same as D but in OFC. **F** Mean decoder accuracy ± SEM
(shading) from an SVM binary decoder trained using balanced neural data from
either M2 or OFC. The decoder was trained to predict whether the choice would be
left or right (*Chosen Side*). We found a main effect of Area and Area x Schedule

interaction using a generalized linear model, with post-hoc analysis revealing better
decoding in M2 than OFC in Schedule 1 ($n_{M2} = 6$, $n_{OFC} = 8$, $p = 0.0027$) and 2
($n_{M2} = 6$, $n_{OFC} = 8$, $p = 0.0438$), but not Schedule 3 ($n_{M2} = 6$, $n_{OFC} = 8$, $p = 0.073$).
Decoder accuracy using shuffled neural data in both M2 and OFC is shown in black.
**G** Similar to (**F**), but here the decoder was trained to predict whether the trial would
be rewarded or not (*Trial Outcome*). We found main effects of Area and Schedule,
but no significant interaction of Area x Schedule. For F and G, the solid line is the
mean, shade is the SEM. *$p < 0.05$. No adjustments were made for multiple com-
parisons. Source data are provided as a **Source Data file**.

For the *Trial Outcome* epoch, we did not find a significant region
difference in the ratio of outcome-selective cells (GLM, $\beta_{Area} = -0.10$,
$p = 0.068$). This ratio was also not modulated by experience alone, as
the session was not a significant predictor. We did, however, find a
significant main effect of schedule (GLM, $\beta_{Schedule} = -0.01$, $p = 0.044$)
and a significant interaction of area and schedule (GLM,
$\beta_{AreaxSchedule} = 0.05$, $p = 0.03$, Supplementary Table 10), resulting from
the fact that the ratio of outcome-predictive neurons in OFC increased
with increasing reward uncertainty (GLM, $\beta_{Schedule} = 0.03$, $p = 0.009$;
Fig. 3E), whereas the ratio of outcome-predictive neurons in M2 was
similar regardless of uncertainty (GLM, $\beta_{Schedule} = -0.015$, $p = 0.44$)
(Supplementary Table 11). It is somewhat paradoxical to observe that
the proportion of neurons encoding trial outcome in M2 remained
stable despite decreases in the certainty of reward. One explanation
for this could be that under conditions of greater uncertainty, addi-
tional M2 neurons are recruited to encode choices.

Prior evidence suggests that value integration may occur at the
time of *Reward Retrieval*[29,40–42], so we performed an additional

analysis to measure the proportion of neurons that were reward-
selective during a 1-s time window centered on head entry into the
reward port during rewarded trials versus a corresponding time
window from unrewarded trials (see "Methods"). There was a larger
proportion of reward-selective cells in M2 compared to OFC (GLM,
$\beta_{Area} = -0.27$, $p = 0.02$) with a significant area by schedule interac-
tion (GLM, $\beta_{AreaXSchedule} = 0.1$, $p = 0.048$) (Supplementary Table 12),
resulting from the fact that OFC exhibited a higher proportion of
reward-selective neurons for schedules with greater uncertainty
(GLM, $\beta_{Schedule} = 0.09$, $p = 0.032$, Fig. 3F), whereas in M2 there was a
similar proportion of reward-selective neurons across all schedules
(GLM, $\beta_{Schedule} = -0.01$, $p = 0.68$). M2 neurons thus appear to be
more heavily recruited overall to encode reward as well as choices
(described above) compared to OFC neurons. Finally, it is an
increase in uncertainty and not experience (Supplementary
Table 13) that drives the increase in the ratio of reward-selective cells
in OFC, since session was once again not a predictor of cell
selectivity.

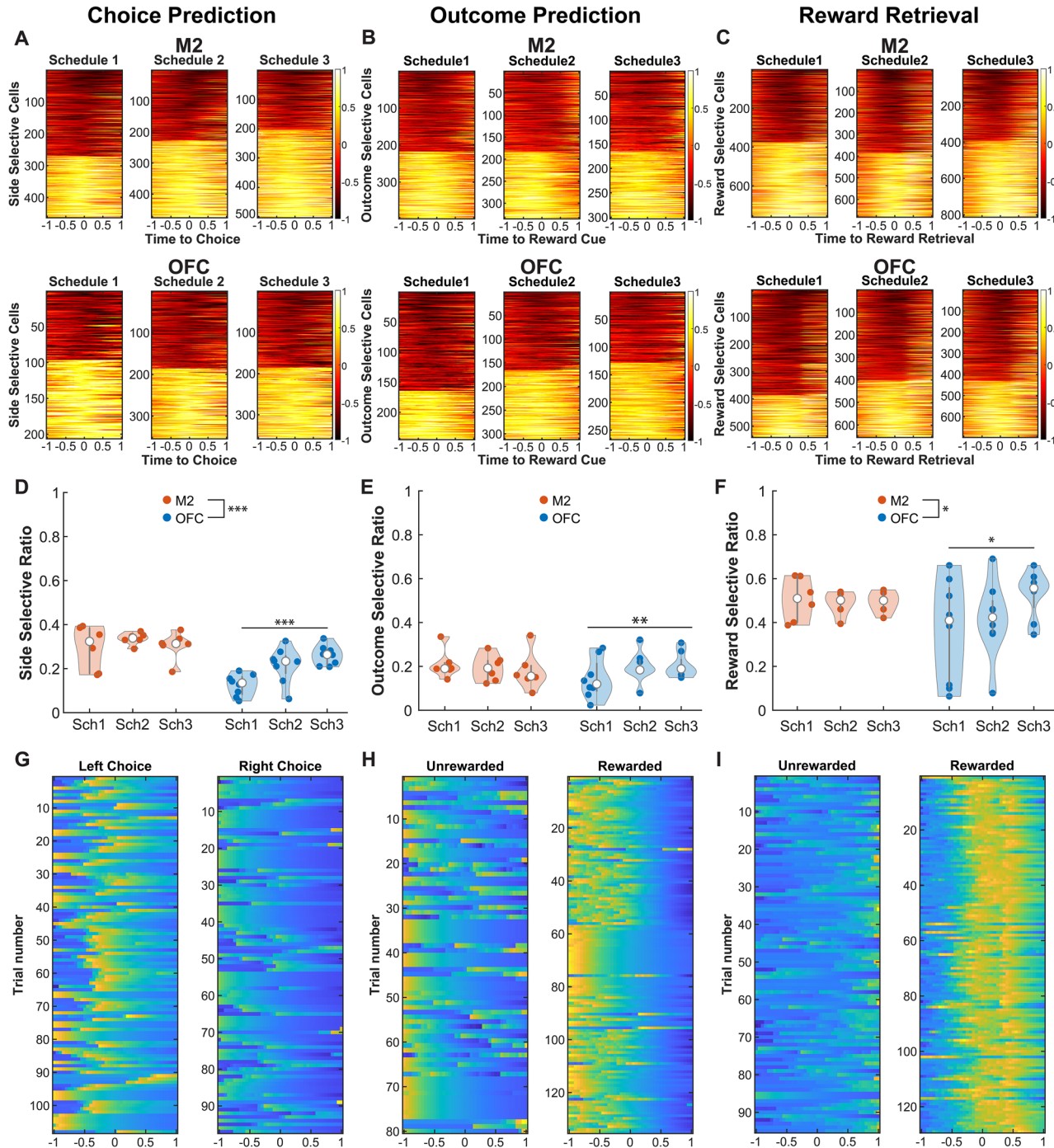

**Fig. 3 | Decoder results are explained by changes in the ratio of selective cells across schedules. A** Each row plots the difference in trial-averaged activity of left trials minus right trials triggered by the time of choice. Cells plotted are all imaged neurons where we observed a significant effect of side, sorted from the most negative beta coefficient to the most positive one. The activity of the neurons was range-normalized. A value closer to 1 indicates more selectivity for the left side trials, and a value closer to −1 indicates more selectivity for the right side. **B**, **C** Same as A, but the difference is the averaged activity of unrewarded trials minus rewarded trials, centered at time of reward cue (**B**) or reward retrieval (**C**). **D** Ratio of side selective neurons is overall larger in M2 ($n_{M2} = 18$, $n_{OFC} = 24$, $p < 10^{-4}$), but it increases with schedule only in OFC ($n_{Sch1} = 8$, $n_{Sch2} = 8$, $n_{Sch3} = 8$, $p = 0.0001$). Group sizes from left to right are: $n = 6$, $n = 6$, $n = 6$, $n = 8$, $n = 8$, $n = 8$. **E** Ratio of

outcome selective neurons increases with uncertainty only in OFC ($n_{Sch1} = 8$, $n_{Sch2} = 8$, $n_{Sch3} = 8$, $p = 0.0091$). Group sizes from left to right are: $n = 6$, $n = 6$, $n = 6$, $n = 8$, $n = 8$, $n = 8$. **F** Ratio of reward selective neurons is overall larger in M2 ($n_{M2} = 18$, $n_{OFC} = 24$, $p = 0.033$) but increases with schedule only in OFC ($n_{Sch1} = 8$, $n_{Sch2} = 8$, $n_{Sch3} = 8$, $p = 0.032$). Group sizes from left to right are: $n = 6$, $n = 6$, $n = 6$, $n = 8$, $n = 8$, $n = 8$. **G** Example of the z-score normalized activity of a significant side-selective M2 neuron on left and right trials. **H** Same as (**G**) but for a significant outcome-selective M2 cell. **I** Same as (**G**), but for a significant reward-selective OFC neuron. The center of each boxplot is the median. All statistical tests performed using a Generalized Linear Model. *$p < 0.05$. **$p < 0.01$ ***$p < 0.001$. Source data are provided as a **Source Data file**.

## Behavioral strategies predict decoding accuracy in OFC, not in M2

To assess the degree to which different behavioral measures (Fig. 4A–F) modulated decoding accuracy in OFC and M2, we analyzed the data using an omnibus mixed-effects GLM for *Chosen Side* (Supplementary Table 14) and *Trial Outcome* (Supplementary Table 16). As predictors, we included primary behavioral measures described above such as median initiation, choice, and reward latencies for the entire session (Fig. 4I–K), as well as secondary measures such as Win-Stay (WS, proportion of trials when the rat repeated a response for which it had been rewarded on the previous trial) and Lose-Shift (LS, proportion of trials when the rat did not repeat a response for which it had been unrewarded on the previous trial), which we and others have previously used to assess adaptive strategies in both stimulus and action-based learning[24,43]. As other predictors, we included a Flexibility Index (see "Methods") that measured the animal's ability to adapt to trial block reversals, and a Perseveration Index, or the mean number of trials to shift choice behavior after being unrewarded, divided by the number of first unrewarded trials[44].

We found significant M2 vs. OFC area interaction effects for Win-Stay, Lose-Shift, and Perseveration Index in the bin just before the choice epoch (Fig. 4H and Supplementary Table 14). Post-hoc GLMs revealed that only initiation latencies (GLM: $\beta_{InitiationLatency} = -0.067$, $p = 0.0324$) significantly predicted *Chosen Side* decoder accuracy for M2 (Fig. 4I and Supplementary Table 15). In contrast, WS (GLM: $\beta_{WinStay} = 0.843$, $p = 0.0008$) and LS (GLM: $\beta_{LoseShift} = 1.466$, $p = 0.0240$) positively predicted, whereas Perseveration Index (GLM: $\beta_{Perserveration\ Index} = -4.128$, $p < 0.0001$) negatively predicted *Chosen Side* decoder accuracy in OFC (Fig. 4J and Supplementary Table 15). Similar post-hoc GLMs (Fig. 4K and Supplementary Table 16) resulted in no significant behavioral predictors for *Trial Outcome* decoding accuracy in M2 (Fig. 4L and Supplementary Table 17), but we again found that WS (GLM: $\beta_{WinStay} = 0.652$, $p = 0.0202$) and LS (GLM: $\beta_{LoseShift} = 1.550$, $p = 0.0228$) positively predicted and Perseveration Index (GLM: $\beta_{Perserveration\ Index} = -1.543$, $p = 0.0498$) negatively predicted *Trial Outcome* decoding accuracy in OFC (Fig. 4M and Supplementary Table 17). These analyses indicate that complex, strategy-level behaviors contribute to decoding accuracy in OFC, but not M2.

## OFC, but not M2, neurons preferentially support adaptive learning under uncertainty

Male and female Long-Evans rats underwent stereotaxic surgery to infuse hM4Di DREADDs in either OFC or M2 (see "Methods"). After following the same pretraining scheme as rats with lens implants, this cohort performed schedules first in order of increasing (i.e., ascending) uncertainty- identical to the cohort with lens implants- and then in order of decreasing uncertainty, which rats with lens implants did not experience. Each schedule was administered twice- with CNO inhibition on the first session and an injection of vehicle (VEH) on the second session- before the rat advanced to the next schedule (Fig. 5A).

As in Fig. 1E, accuracy was measured as the raw choice (0,1) of the higher reward probability option (see "Methods"). Using omnibus GLMs with binomial distribution (Supplementary Tables 18 and 19), we assessed if there were interactions between schedule (1, 2, 3), order (ascending or descending uncertainty), and drug (CNO, VEH). OFC inhibition impaired accuracy across all schedules, as the effect of the drug did not interact with schedule or order (CNO: $\beta_{Drug} = -0.2831$, $p < 0.02$), Supplementary Table 18. In contrast, while there was also an effect of inhibition of M2 on accuracy (CNO: $\beta_{Drug} = -0.4698$, $p < 0.0001$), there was a drug by schedule interaction (Supplementary Table 19), which revealed its effect was in Schedule 1 (CNO: $\beta_{Drug} = -0.2018$, $p < 0.0001$) (Supplementary Table 20). Due to an interaction of drug by order, we conducted a post-hoc analysis which revealed that decreased performance following M2 inhibition was in the ascending (CNO: $\beta_{Drug} = -0.0950$, $p = 0.0176$) but not descending

order (Supplementary Table 21). Consistent with prior findings, OFC neurons support first reversal experience[24,31,45] as well as learning under all subsequent uncertainty levels. In contrast, the largest effect size for M2 neuron involvement during learning occurred in Schedule 1.

We also examined the effect of inhibition on adaptive strategies (Supplementary Tables 22–25). OFC inhibition reduced the Win-Stay strategy across all schedules (CNO: $\beta_{Drug\ CNO} = -0.020$, $p < 0.01$), with no effect on Lose-Shift. M2 inhibition had no significant effect on the use of either strategy (Fig. 5B). Finally, we analyzed initiation, choice, and reward latencies following inhibition. While M2 inhibition had no effect on choice latencies (Supplementary Table 26), OFC inhibition significantly slowed choice latencies (Supplementary Table 27 and Fig. 5F). All analyses included only animals with confirmed, bilateral expression (Fig. 5D, E).

## Discussion

We imaged single-cell activity in M2 and OFC while freely-moving rats underwent learning. Using a binary support vector machine decoder, we found evidence of functional heterogeneity in the frontal cortex at both the population- and single-cell level. Specifically, M2 consistently contained more information about choice on individual trials than OFC, yet decoder accuracy increased with uncertainty, not just experience, only in OFC, not M2. We found that decoder accuracy was predicted by adaptive strategies in OFC, but not M2. We also determined that M2 and OFC neurons exhibited different neural trajectories in learning trial types: these changed dynamically in more uncertain schedules in OFC, but not M2 (Supplementary Fig. 4). Our single-cell analyses revealed a recruitment of an increasing ratio of choice-, outcome- and reward- selective neurons in OFC with uncertainty, not M2. Finally, chemogenetic inactivation of each of these regions confirmed functional roles for OFC, but not M2, as preferentially involved in learning under uncertainty. Collectively, these results suggest that neurons in OFC are more engaged in conditions with greater task demands that require adaptive strategies, such as increased uncertainty during learning.

There are several significant extensions and confirmatory findings here. First, studies of neural activity in the frontal cortex in mice[46], rats[47,48], and especially non-human primate subjects[49] typically involve extensive training prior to surgery to ensure that the animal is a task expert prior to any recording or imaging. In contrast, in the present experiment, rats learned the task de novo while we imaged single cells from these regions, which allowed us to determine that the dynamics in OFC changed as the uncertainty increased. Quite differently, M2 signaling of choice was robust and stable throughout the experiment. With the exception of two single-unit electrophysiological studies[11,12], these two brain regions have not been systematically compared for their signaling in a reward-motivated task until now, and importantly, the previous studies administered a dynamic foraging task where subjects were not required to learn any task structure. We used a similar task to that in a recent report by Hattori and colleagues[39], but those authors also did not vary the uncertainty in the reward environment, as we did here. Consistent with their findings, we confirm that OFC plasticity is required for across-session learning (what the authors call "meta-learning"), and here we extend this to conditions of increasing uncertainty encoded in OFC, but not in M2, neurons.

Studies following manipulations in rat OFC in flexible reward learning have included targeting the entire ventral surface[50,51] or comparisons of medial vs. lateral OFC[27,28]. Similar to a recent study using identical targeting[24], here we examined the role of ventrolateral OFC, a subregion not as often probed as medial and more lateral OFC[52], and found evidence consistent with its encoding of expected uncertainty[25]. Specifically, several studies using different methods[32–34,53] suggest OFC must be 'online' to establish a baseline risk

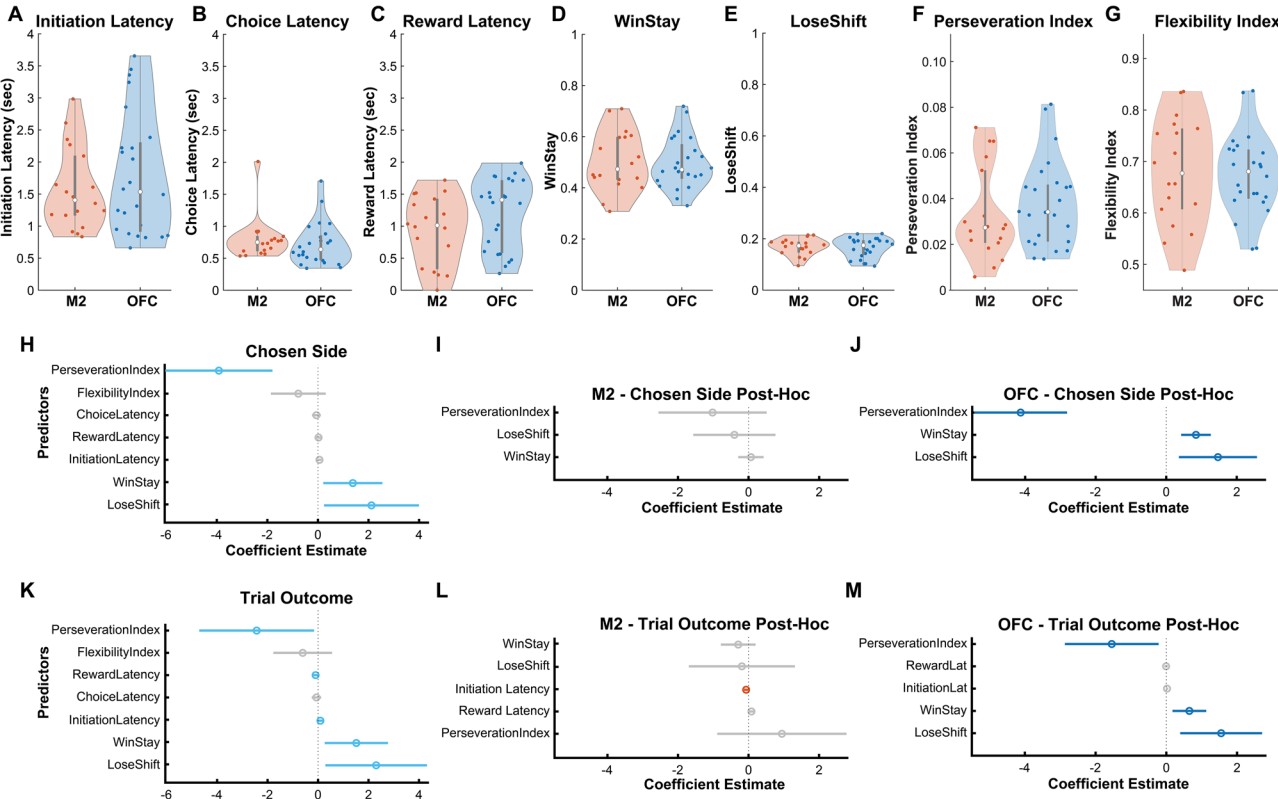

**Fig. 4 | Chosen Side and Trial Outcome decoding accuracy is predicted by behavioral strategy only in OFC. A–G** Behavioral measurements that were entered as regressors in omnibus Generalized Linear Models (GLM). The group sizes were the same across all measurements: $n_{M2} = 18$, $n_{OFC} = 24$. The center of the boxplot is the median. Latency measures (**A–C**) were calculated as medians of all trials for each session. WinStay and LoseShift (**D, E**) were calculated as the proportion of total trials that were WinStay and LoseShift at the session level. Perseveration Index and Flexibility Index (**F, G**) were calculated using formulas described in the Methods. **H** Beta coefficients for regressors where significant area (M2 vs. OFC) interactions were found in predicting decoder accuracy of Chosen Side. Modeling was performed only on the accuracy from the bin right before

choice (highlighted in yellow, Fig. 2). **I** Post-Hoc GLM performed on sessions from rats with lenses in M2. **J** Post-Hoc GLM performed on sessions from rats with lenses in OFC. Strategy indices were significant predictors of Chosen Side decoding accuracy in OFC, but not in M2. **K** Beta coefficients for regressors where significant area (M2 vs. OFC) interactions were found in predicting Trial Outcome. Similarly to (**H**), analyses were only performed on the bin before the reward cue (highlighted in yellow, Fig. 2). **L** Post-Hoc GLM performed on sessions from rats with lenses in M2. **M** Post-Hoc GLM performed on sessions from rats with lenses in OFC. Strategy indices were significant predictors of Trial Outcome decoding accuracy in OFC, but not in M2. Source data are provided as a **Source Data file**.

associated with probabilistic outcomes from which to gauge meaningful changes (i.e., reversals or block switches). OFC-inhibited animals also do not employ effective Win-Stay strategies[24], consistent with the GLM modeling results of the imaging experiment and the causal chemogenetic perturbation result here. A limitation in the present experimental design is that we inactivated OFC or M2 neurons on the first experience with each schedule (i.e., CNO was administered on the first and VEH was administered on the second of two sessions of each schedule). While there is converging evidence that OFC is most involved in first reversals[24,31,45] we were mainly interested in the pattern across schedules, and across M2 vs. OFC, not within a schedule. Indeed, we found the pattern of learning attenuations was different across schedules, and the two regions supported different facets of learning.

We also find evidence that OFC neurons learn task structure details that are ultimately useful in predicting the consequences of decisions in uncertain and/or novel conditions[15]. In support of this, we found that OFC neurons do not encode choices as well as M2 neurons during more certain schedules, evidenced in both decoder and stability analyses, where M2 decoder accuracy is greater than that of OFC in the more certain schedules. In addition, we observed an increase in the proportion of neurons that signaled reward with increasing uncertainty in OFC, not in M2. These findings are consistent with the idea that OFC has a pivotal role in either using or creating a task map[54,55]:

just as a spatial map helps one move better through space, a task map would help with adaptive behavioral strategies. We show evidence of this with decoder performance in OFC positively predicted by adaptive strategies WinStay, LoseShift, and negatively correlated with an inflexibility measure like Perseveration Index.

By comparison to OFC, M2 is often overlooked in learning studies due to its presumed stronger role in motor skill and action execution in both rodents and non-human primates[56–58]. There is evidence that M2 encodes task-relevant features, such as whether a rat will choose the left or right side in a bandit task[12] and a risky or safe option[59]. Thus, our results provide additional evidence that M2 encodes important task parameters such as choice and trial outcome. Indeed, M2 is relatively unchanging: its encoding of choice remained robust and dynamics in this region contained a full picture of task-relevant features even from the initial phases of learning. In support of this, we found that M2 contained information related to blocks throughout all different epochs of a trial, more so than OFC (Supplementary Fig. 5). In addition, chemogenetic inactivation of M2 preferentially impacts learning in the most certain condition (Schedule 1). Given that M2 biases decision-making processes[5,10], a read-out of "intention" could be a useful brain-machine interface application, given its stable coding.

These findings demonstrate distinct roles for M2 and OFC in processing task-specific information during learning under uncertainty, with M2 neurons maintaining robust, stable activity throughout

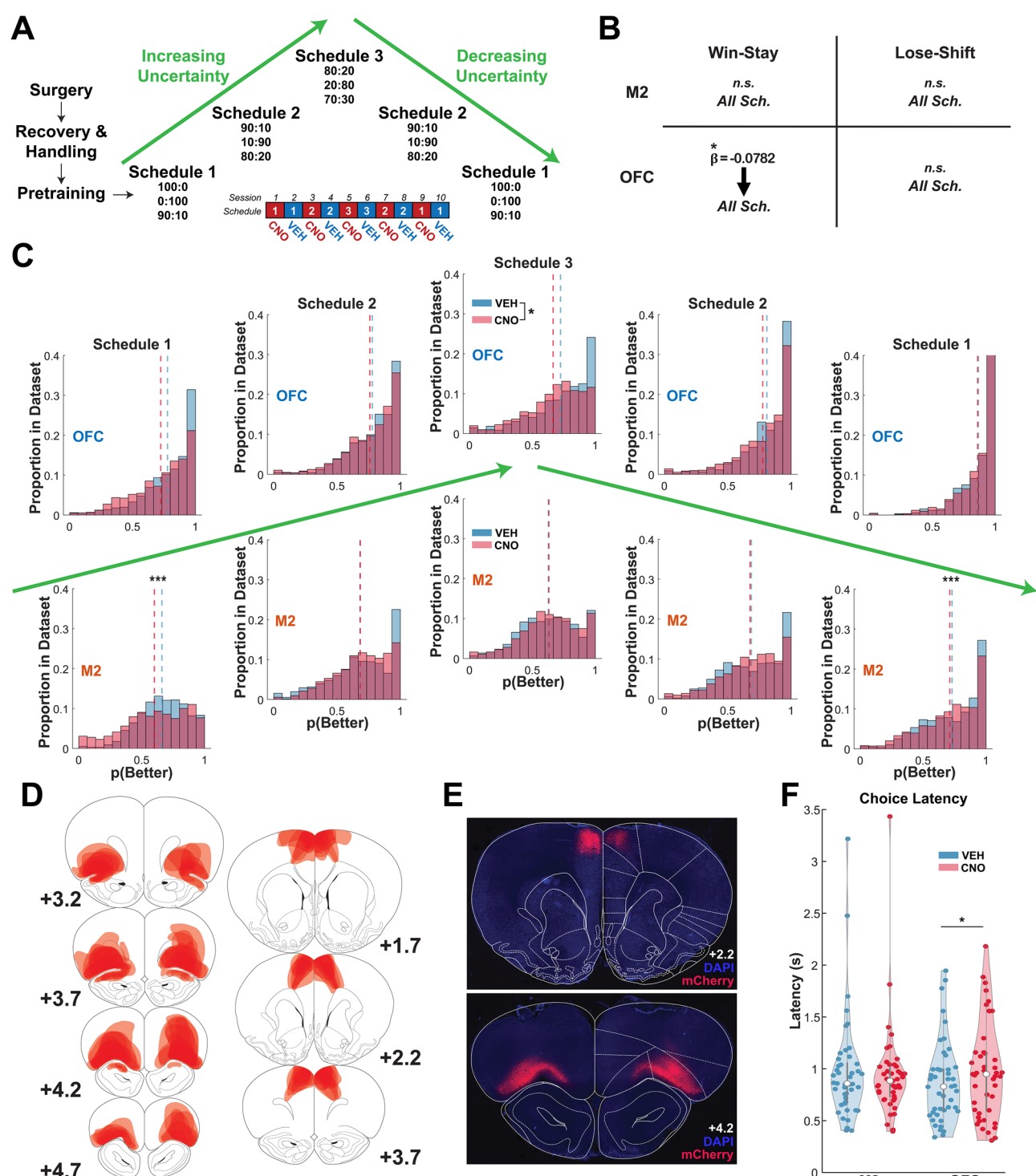

**Fig. 5 | OFC, but not M2, neurons support learning under all uncertainty levels.**
**A** Following surgery, recovery, handling, and pretraining rats were introduced to different schedules, first in increasing order of uncertainty followed by decreasing (reverse) order. Each schedule was administered across two days, the first under CNO inhibition and the second following an injection of vehicle (VEH). **B** Win-stay behavior was reduced across all schedules following OFC inhibition, but not M2. **C** OFC inhibition resulted in attenuated accuracy throughout learning ($n_{VEH} = 10125$, $n_{CNO} = 10125$, $p = 0.018$). In contrast, M2 inhibition impaired only Schedule 1

performance ($n_{VEH} = 4050$, $n_{CNO} = 4050$, $p < 0.0001$). **D** Reconstructions of DREADD expression across all rats. **E** Representative photomicrographs of hM4Di-mCherry DREADD expression in M2 and OFC. Coronal sections reprinted from *The rat brain in stereotaxic coordinates*, 7th edition, Paxinos, G. & Watson, C., 2014, with permission from Elsevier. **F** OFC ($n_{VEH} = 45$, $n_{CNO} = 45$, $p < 0.0001$), but not M2, inhibition slowed choice latencies. All statistical analysis were performed using a Generalized Linear Model. $*p < 0.05$ $***p < 0.001$. Source data are provided as a **Source Data file**.

learning, and OFC neurons exhibiting significant adaptations in response to uncertainty. This suggests a mechanism by which OFC neurons make adjustments to dynamic environments, while M2 provides a more consistent response control mechanism. Interestingly,

unlike the present findings that suggest a role for rodent OFC in both outcome and reward probability encoding, primate OFC is not as important for encoding of reward "availability, as indexed by its probability"[60]. Instead, OFC is more specialized in primates for value

and "desirability," not probability. This may be an important primate-rodent difference to consider when interpreting experimental findings in the frontal cortex across species in the future.

## Methods

### Subjects

A total of $N = 32$ Long-Evans rats was used. Subjects for single-cell calcium imaging were $n = 7$ adult Long-Evans rats: $n = 4$ (3 males, 1 female) with lenses in OFC and $n = 3$ (2 females, 1 male) with lenses in M2. Subjects for chemogenetic manipulation were $N = 25$ adult Long-Evans rats obtained from Charles River Laboratories. Of these, seven animals were excluded from analyses: one due to a failure to complete pretraining and six due to weak or unilateral viral expression. Thus, the total for the chemogenetic experiment was $N = 18$: OFC: $n = 9$, 5 females; M2: $n = 9$, 5 females. Rats for calcium imaging were obtained from a colony of GAD-Cre transgenic rats (RRRC#751; Rat Resource and Research Center, MO) / TH-Cre (RRRC#659; Rat Resource and Research Center, MO), genotyped as wildtype. Rats were pair-housed when they arrived at our housing facilities and then single-housed after undergoing surgery. Rats were housed in plastic transparent tubs filled with bedding media made of wood shavings in a room with a reverse 12 h light cycle. Pretraining and testing occurred during their dark period, seven days per week, until all experimental sessions were completed. Motivation was achieved through a progressive food restriction until the weight was kept above 85% of their initial free-feeding weight. Rats had free access to water at all times. Before training, rats were handled for 10–20 min for a minimum of 5 consecutive days. Subsequently, rats were introduced to a three-finger hold that enabled intraperitoneal injections or the attachment of a miniscope after baseplating. Experiments were conducted with the approval of, and following the guidelines established by, the Chancellor's Animal Research Committee at UCLA.

### Behavioral pretraining

Our operant chambers were outfitted with a touchscreen on one end, divided into three zones (left, center, and right), and a reward port opposing the touchscreen (Models 80604 and 80604 A, Lafayette Instrument, Inc., Lafayette, IN). Ten sucrose pellets were placed in the rat's homecage one day before training to acclimate them to the pellets. The rats were placed in the operant chambers on the first training day, and 5 pellets were dispensed. The rats remained in the chambers for 20 min, where they could explore, eat the pellets, and become habituated to the testing environment. The criterion for the rat to move on to the next phase is eating all 5 pellets within the allotted time. The following day, they begin Initial Touch Center (ITC), which was the first time the rat had learned to interact with the touchscreen. A trial commenced with the display of a white rectangle in the center zone to indicate the rat is required to interact with it (e.g., nose poke or paw touch). Once the rat interacted with the center zone, a pellet was immediately delivered to the reward port. The rat only had 40 s to interact with the center zone before the rectangle disappeared, and the trial was counted as an initiation omission. If the rat successfully interacted with the center zone and collected the sucrose pellet, the trial was considered a committed trial. For the rat to move on from ITC, it had to complete 60 committed trials within 45 min. The final phase of pretraining was Immediate Reward (IR). In this phase, the rat was required to initiate the trial by touching the center zone when the white rectangle is presented, and then it was required to touch a stimulus presented pseudo-randomly either on the left or right zone of the touchscreen. The rat had 60 s to interact with this stimulus to obtain a reward. Similar to the ITC criterion, for the rat to complete IR, it had to complete 60 committed trials within 45 min. After completing IR, rats were considered ready for surgery.

### Miniscope assembly and imaging

We purchased assembled Miniscope V4.4 kits from *Open Ephys*. The miniscope was placed in a 3D-printed, custom-designed, and constructed helmet (a link to the 3D design would be excellent here). The 3D-printed helmet enabled us to utilize a more resilient SMA connector for rat movement. Once the miniscope was fitted inside the helmet, we ensured LED power, the electrowetting lens, and the CMOS were working correctly. The tether used to connect the miniscope to the operating system had a female SMA connector on one end and a male HDMI connector on the opposite end. A motorized commutator from Doric Lenses (Part Number: AERJ_24_HDMI + 4) allowed the tether to rotate with the rat. The commutator transferred the signal to the miniscope DAQ v3.3, also obtained from *Open Ephys*. The DAQ sent the signal over USB to a recording computer. Additionally, the DAQ produced a TTL pulse every time the miniscope recorded a frame of data. These TTL pulses were recorded in parallel with TTL pulses sent by the operant chamber whenever certain behaviors within a trial were performed (i.e., trial initiation, reward port entry) and were used to time-lock neural data to specific epochs within trials and within single sessions. These TTL pulses were recorded using a CerePlex Direct data acquisition system and later used to sync the neural data. The USB signal from the miniscope was recorded onto a Framework Laptop 13 (11th Gen Intel) using the Miniscope DAQ Software (build v1_11). All imaging was conducted at a rate of 20 frames per second (fps).

### GCaMP and lens implant surgeries

Rats were anesthetized with isoflurane (5% induction, 2% maintenance in 2.5 L/min $O_2$) and placed in a stereotaxic apparatus (Kopf Instruments, Tujunga, CA). After clearing the skull, we leveled the skull by ensuring the difference in DV around the bregma was not greater than 0.05 mm. An etchant was applied to the skull for 1 min to ensure the adhesive would adhere. After cleaning the etchant, we partially drilled seven tiny holes that would be used for seven anchor screws (Fine Science Tools #19010-00). Two anchor screws were placed in the very front, two directly posterior to the bregma, two anterior to the lambda, and one contralateral to where the GRIN lens would be placed. Subsequently, a 1.2 mm diameter craniotomy was drilled above the brain regions of interest. For OFC placement, the coordinates were AP: + 3.7 mm, ML: + 2.5 mm, and for M2 placement, the coordinates were AP: + 2.0 mm, ML: + 0.7 mm. The craniotomy was cleaned to ensure that the superior sagittal sinus was intact, and then the dura was carefully removed. A 1 μL Hamilton syringe and pump delivered GCaMP6f (AVV9-CamKII-GCaMP6f-WPRE-SV40, Addgene, # 100834) through cannulae and injectors in either OFC or M2. For OFC surgeries, we unilaterally infused virus into two different sites (AP: + 3.7 mm, ML: + 2.5 mm, DV: − 4.2 mm and − 4.6 mm). For M2, we also infused unilaterally into two different sites (AP: + 2.0 mm, ML: + 0.7 mm, DV: − 1.4 mm and − 0.9 mm). We dialed down 0.05 mm more ventrally in either region to create a pocket for infusions. We then injected 0.5 μL of GCaMP6f per site, thus totaling 1 μL of virus per surgery. The syringe was placed in a Hamilton pump with a speed of 0.1 μL per minute. Following each infusion, 5 min elapsed before the cannula was moved. Lens implantation occurred immediately after viral infusions. A 23-gauge needle was attached to the stereotaxic instrument, zeroed at bregma, and then lowered to make clear a path for the lens. Next, the lens was lowered to the halfway point between the DV coordinates as above using a vacuum. After the GRIN (Gradient Index) lens had been placed in its location, we used viscous Loctite super glue to glue the sides of the lens to the top of a dry skull, then added a layer of Metabond bone cement that coats the lens, anchor screws, and the most superficial layer of the skull. After allowing 5–10 min for the Metabond to dry, we poured opaque dental dement on the Metabond to fully cover the anchor screws and cover the sides of (but not obscure) the GRIN lens. Before the dental cement thoroughly dried, we affixed a custom 3D-printed lens cover. After drying the dental cement, we

applied a layer of topical antibiotic to the edge of the dental cement head cap. Animals were administered carprofen (5 mg/kg s.c.) and saline (1cc) prior to completion of surgery. Post-operative care consisted of carprofen (5 mg/kg s.c.) for five days following surgery.

We waited 3 to 4 weeks post-surgery to allow for GCaMP expression before attaching the baseplate. We anesthetized the rat for baseplating, removed the lens protector, and cleaned the top of the dental cement head cap. We attached a baseplate to a fully assembled miniscope using a custom 3D holder. We lowered the miniscope and the baseplate until we could focus entirely on the top of the implanted GRIN lens, searching for visible expression in the area of interest, then fixing the baseplate using dental cement. GCaMP expression and lens implant reconstructions (Adobe Creative Cloud) on coronal sections[61] are shown for OFC (Fig. 1I, **left**) and for M2 (Fig. 1I, **right**), along with representative photomicrographs (Fig. 1J, K). All GCaMP and lens placements for all animals are shown in Supplementary Fig. 1.

### Viral surgeries
Craniotomies were performed similarly to the GCaMP surgeries. To express inhibitory hM4Di DREADDs (Designer Receptors Exclusively Activated by Designer Drug) in cortical excitatory neurons, a CaMKIIα promoter drove the hM4Di-mCherry sequence (Addgene viral prep # 50477-AAV8, Watertown, MA). This construct was infused bilaterally into either OFC or M2 at the same coordinates as previously described. 0.5 μL was infused per site, thus totaling 1 μL per hemisphere and 2 μL per surgery. The scalp was carefully closed using self-dissolving sutures to ensure a fast and less invasive recovery.

Experimental testing began a minimum of three weeks after surgery to allow for receptor expression. Reconstructions of DREADD expression are shown for OFC (Fig. 5D, **left**) and for M2 (Fig. 5D, **right**), along with representative photomicrographs (Fig. 5E). Reconstructions were generated using commercial software (Adobe Creative Cloud) on coronal sections from Paxinos & Watson[61].

### Behavioral testing
After the baseplating surgery, rats were again put on food restriction before behavioral testing: they were eased into a restricted diet over three days to no more than 85% of their free-feeding weight. In addition, rats were once again habituated to a gentle hold and plugging the miniscope into the baseplate. Rats were given a second round of pretraining as an acclimating imaging session using a miniscope. We ensured rats performed many trials in the IR phase (> 200 trials in 45 mins) before moving the rat onto the main behavioral task.

*Learning of Probabilistic Reversals.* Similar to pretraining, a trial began when a circle was displayed in the center zone of the touchscreen. The rat must interact with it by nose-poking or touching it with its paw within 40 s; otherwise, it is scored as an initiation omission. After touching the circle, two identical stimuli were presented on both the right and left touchscreen zones. The rat had to interact with either the left or the right zones. The rat had 60 s to make a choice otherwise, it was scored as a choice omission. One of the zones was associated with a higher probability of reward, while the other was associated with a lower probability of reward. If the rat received a sucrose pellet reward, it was dispensed in the reward port on the opposite side of the chamber, and the subsequent trial began after a 10 s inter-trial interval (ITI). If the rat did not receive a reward, a 5 s time-out was imposed in addition to the 10 s ITI. If the rat initiated a trial and made a choice within the allotted time, the trial was scored as a committed trial regardless of whether it was rewarded. Food motivation was assessed purely by the number of committed trials, not accuracy. In each session, there were three blocks of 75 committed trials each.

From one block to the next, the location of the highest-valued side reversed. The probability of each option itself varied by block according to schedule, with Schedule 1 including deterministic blocks (100:0, 0:100, 90:10) and Schedules 2 (90:10, 10:90, 80:20) and 3

(20:80, 80:20, 70:30) being entirely probabilistic. There was an increase in uncertainty over testing. There were two primary reasons for this design: (1) reversals (i.e., block design) keep the animals engaged and less likely to fall into a motor habit of always responding left, for example, and (2) we enhance the chance of including the same cell population during learning at one level of uncertainty (e.g., 100-0) with another (90-10), if these blocks occur in the same session. Task structure is visualized in Fig. 1A. Violin plots for all initiation, choice, and reward latencies are shown by schedule (Fig. 1F–H), and the median per session collapsed across all schedules for each of these groups.

*Chemogenetic inhibition.* In the chemogenetic cohort (n = 18), inhibition of OFC or M2 was achieved with systemic administration of clozapine-N-oxide, CNO (6 mg/kg, i.p., in 95% saline, 5% DMSO; HelloBio, Princeton, NJ) 30 min prior to behavioral testing. To control for the nonspecific effects of injection and handling, vehicle solution, VEH (95% saline, 5% DMSO), was alternatively administered. The effects of CNO were assessed in a within-subjects design where rats completed each schedule twice, first under CNO administration and second with an injection of VEH. To investigate learning in changing environments while controlling for the effect of task experience in navigating the task, rats first completed the schedules in increasing order of uncertainty, then repeated these schedules in decreasing order (Fig. 5A).

### Histology
Following completion of behavioral testing, all rats were euthanized with an intraperitoneal dose of sodium pentobarbital and sodium phenytoin (Euthasol® Euthanasia Solution) and transcardially perfused with PBS and a 4% formalin solution. Brains were extracted, placed in 4% formalin solution for at least 24 h, and subsequently left in a 30% sucrose solution for at least 24 h. Forty μm coronal sections were mounted onto slides and coverslipped with DAPI to visualize GCaMP or hM4Di-mCherry expression. Imaging was performed using an LSM 900 confocal microscope (Zeiss) and a BZ-X710 florescence microscope (Keyence).

### Post-processing calcium imaging pipeline
Miniscope recording videos were processed using custom Python code and the Python version of CaImAn[62]. We concatenated videos and cropped them to focus the field of view (FOV) with the neural activity. We then passed it through the motion correction algorithm in CaImAn, with the central parameter *gSif_filt* set between 6 and 12 for our data. All of the motion correction for our data was conducted with the piecewise rigid motion correction Boolean setting set to true. We performed the non-rigid algorithm if this algorithm did not fix the motion artifacts. Once the motion correction was completed, the motion-corrected videos were passed on to the CNMF-E portion of CaImAn[62]. The main parameters we used for the CNMF-E were as follows: the half size of a neuron = 8; the half size of the neuron's bounding box = 16; for every session, we chose a value to separate the first and most significant population of minimum peak-to-noise (PNR) ratio resulting in use of a range between 7–16; for the minimum correlation value we found values between 0.75 and 0.85 worked best. Every other setting was left to the CaImAn default. After CNMF-E, we performed a component analysis. We set the Boolean parameter *use_cnn* to true so that CaImAn would use a trained convolution-neural network (CNN) model to assign a value between 0-1 that determined how close the components matched the appearance of soma. The algorithm used three different values to determine if a component should be as accepted as a cell: Spatial footprint consistency (*rval*), Signal-to-Noise ratio (*snr*), and CNN threshold value (*cnn_thr*), with the minimum values for each of these set to 2.5, 0.85, and 0.85, respectively. We then visually inspected random components. Since, according to CaImAn documentation, it is not advised to calculate deltaF/F when using 1P data (due to the inability to confidently

calculate baseline with background activity), we used the function *estimates.detrend_df_f* with the flag *detrend_only* set to true to calculate fluorescence where baseline has been subtracted but has not been normalized by it. Additional settings used were: quantileMin = 8, frames_window = 250, use_residuals = True. Finally, we calculated the deconvolved traces of the accepted components using the CaImAn function *estimates.deconvolve*. This removed any change in the fluorescence not due to the dynamics of the neurons. After the detrended and deconvolved values were calculated, all of the outputs were saved into a.mat file (MATLAB, MathWorks Inc., Natick, MA), for all further processing. In MATLAB, we combined the output from CaImAn with TTL pulses to align the activity of neurons around specific epochs of behavior.

Some of the videos had banding artefact caused by a known issue with the USB controller of the UCLA miniscope DAQ. When banding occurred, we first saved the frames in which this occurred and then filled the frames with the last reliable frame before the banding. Following processing with CaImAn, we replaced the data from those frames with NaNs.

### Data Analysis

All data analyses were conducted using custom-written code in MATLAB R2023b and Python version 3.10.

*Learning and Performance*. The probability of choosing the better option was the primary measure of interest, and secondarily Initiation Latency, Choice Latency, and Reward Latency were each analyzed with a series of mixed-effects General Linear Models (GLMs) (*fitglme* function; Statistics and Machine Learning Toolbox) first with omnibus analyses that included lens group (M2-implanted, OFC-implanted). Follow-up GLMs were conducted pending significant interactions in the omnibus models. The coding of variables for GLM were: 0 = females, 1 = males, 0 = OFC-implanted, and 1 = M2-implanted). Statistical significance for GLM analyses was noted as p-values of less than 0.05. The variable of interest p(Better) was calculated starting with a boolean vector where a 1 corresponded to a choice for the better option: 1 with the higher probability of reward and a 0 for the worse option. This vector of raw choices (0, 1) was then analyzed using a GLM with a binomial distribution. The Win-Stay and Lose-Shift values presented in Fig. 5B and Supplementary Tables 22–25 were calculated similarly to p(Better), but for the session, not trial-by-trial. A boolean vector was generated for the entire session: 1 if the trial was Win-Stay / Lose-Shift and 0 if it was not. These boolean vectors were then input into *movmean* to generate a proportion of trials within this window that were Win-Stay or Lose-Shift. The following are definitions of predictors: *TrialNum*: The number of trials within an entire session (from 1 to 255); *Session*: Experimental day (from 1–6 in the calcium imaging and 1–10 in the DREADDs experiment); *Block*: The number of blocks within a single session (from 1–3); *Area*: The brain area for calcium imaging (1 = M2, 0 = OFC); *Schedule*: The uncertainty schedule (from 1–3). The remaining predictors are defined in the Results section.

*Cross Validated Peak-Aligned Heatmap*. A window of detrended fluorescence 1 s before and 3 s after choice was obtained for every trial in a session. This resulted in a matrix of cell-by-frame-by-trial averaged across all trials for a resulting cell-by-frame matrix. This was normalized using the normalize function on MATLAB with the 'range' parameter. The max in a cell's activity was 1, and the minimum was 0 to ensure fair comparison across all cells in that session. The odd trials of neurons were then peak aligned, and the resulting indices were used to plot the even trials using *imagesc*. Vertical dashed bars show the moment of choice and the moment of reward cue.

*Decoder Analysis*. A 4 s window of detrended fluorescence centered around Initiation was obtained for every trial in a session. Separately, another 4 s window (1 s before and 3 s after) of detrended fluorescence around choice was obtained for every trial in a session. This resulted in a cell-by-frame-by-trial matrix for initiation and a

separate one for choice. We temporally downsampled so that activity in bin 1 was the average of frames 1–6, activity in bin 2 was the average of frames 3–9, and so on. Every trial was also labeled as either a left or right choice, whether rewarded or unrewarded (win or lose). One trial was left out for testing the decoder, and the rest of the trials were then balanced so that there would be the same number of left-rewarded, left-unrewarded, right-rewarded, and right-unrewarded conditions across all sessions in training of the decoder. The balanced trials were randomly selected for every test trial to sample the entire dataset of trials per session. Once the balanced subset of trials was selected, the neural data for that session and the labels were inputted into the Matlab function *fitcsvm* with the "KernelFunction" set to 'linear.' The decoder was then tested on the neural data for the test trial, and the result was tallied as a 1 if the label matched the test label and a 0 if it was mismatched. This was conducted for every single trial in a session. In addition, we reran the decoder on all trials per session, but with the trial type labels randomly shifted. This data is shown as "Shuffled" in Fig. 2F, G. Statistical comparisons and post-hocs were performed using Matlab's *fitglme* function, and the formulas are included in the corresponding tables. Accuracy measures from the bin (average of 5 frames) before choice were utilized for the *Chosen Side* GLMs, and accuracy measures before the reward cue were used for the *Trial Outcome* GLM.

*Single Cell Selectivity GLM*. Activity for every neuron was obtained on the same bins used for the GLMs for decoder accuracy. In addition to the *Chosen Side* and *Trial Outcome* activity, we used the averaged activity 5 frames before and after *Reward Retrieval* for single-cell selectivity analysis. Statistics were conducted using Matlab's *fitglme* and the formulas for $\gamma$ = neural activity were $\gamma$ ~ [1 + side] for Chosen Side and $\gamma$ ~ [1 + rewarded + side] for both Trial Outcome and Reward Retrieval. We added side as a covariate to remove the influence of side-selective neurons appearing as Trial Outcome or Reward Retrieval selective neurons. After a GLM was fitted for every cell per session, the ratio of selective cells (ones with a p-value lower than 0.05) was obtained. We then fit GLM models to the ratios across both areas and all three schedules. The Beta coefficients used for Supplementary Fig. 3 were obtained from the same GLMs used to obtain these ratios. They are also from the same time bins as Fig. 2F, G and the Reward Retrieval timepoint described above.

*Behavioral measurements*. Latencies were recorded from ABET in Lafayette Instrument chambers. Initiation latency measured the time from the presentation of the white circle on the center touch zone until a rat interacts with the middle zone. Choice latency measured the time from the presentation of both stimuli to the time the rat interacts with one of the stimuli to make a choice. Reward latency measures the time from the dispensed pellet until the rat enters the reward port to retrieve the pellet. WinStay was calculated as the proportion of all trials when the rat received a reward on the previous trial and chose the same side for the following trial. LoseShift was calculated as the proportion of all trials when the rat did not receive a reward and chose a different side for the subsequent trial. The Perseveration Index was calculated as the average number of trials to change choice after being unrewarded. Flexibility Index was calculated as follows:

$$FlexIndex = mean\big[(mean(C_{Better}Block1, C_{Better}Block2), (mean(C_{Better}Block2, C_{Better}Block3)\big]$$

where $C_{Better}$ is equal to 1 if the animal chose the better option and 0 otherwise, with the better option defined according to the reward schedule for each block. Statistical comparisons and post-hocs were performed using Matlab's *fitglme* function, and the formulas are included in the corresponding tables. Accuracy measures from the bin before choice were utilized for the *Chosen Side* GLMs, and accuracy measures from the bin before the reward cue were used for the *Trial Outcome* GLM.

## Statistics and Reproducibility

We implemented the following: (1) inclusion of a vehicle control group and random assignment of rats into drug groups for the chemogenetic experiment; (2) data analysis involved all rats and cells; exclusion occurred only if AAV injection sites and/or lenses were mistargeted; and (3) epochs of interest for behavior and imaging were defined before the start of the experiment. Though the collection of behavioral data was not blinded, the methods for collection were completely automated. Generalized Linear Models (GLM) were used to analyze most data, and the formulas were determined a priori to reflect the experimental design. The estimated number of animals was based on our lab's previous experience with the same techniques and behavior. The data for the imaging and chemogenetic experiments were obtained in two separate cohorts each, for a total of four cohorts of animals.

## Reporting summary

Further information on research design is available in the Nature Portfolio Reporting Summary linked to this article.

## Data availability

Data are also available in the following repository https://gin.g-node.org/aizquie Source data are provided in this paper.

## Code availability

Custom codes are available in the following GitHub repository: https://github.com/izquierdolab.

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

## Acknowledgements

This work was supported by NIH 2R01DA047870 (A.I.), NSF NeuroNex 1707408 (H.T.B.), and NIH F31MH135698 (J.L.R.S.). We thank the members of the Izquierdo and Wikenheiser labs for their feedback and suggestions. We would also like to thank Dr. Evan Hart for technical guidance and training on miniscopes, and members of Romero Sosa's dissertation committee, Drs. Kate Wassum and Dean Buonomano, for their feedback on experimental design. Finally, we thank Dr. Dmitriy Lisitsyn for help setting up the CaImAn pipeline and Dr. Alireza Soltani for his valuable feedback on analyses and interpretation.

## Author contributions

A.I.: Conceptualization, methodology, formal analysis, resources, writing-original draft, writing-review & editing, supervision, and funding. J.L.R.S.: Conceptualization, methodology, formal analysis, investigation, data curation, writing- original draft, writing- review & editing, software, and visualization. A.Y.: Investigation, data curation, formal analysis, and visualization. A.M.W.: Formal analysis and writing- review & editing. H.T.B.: Formal analysis, resources, writing- review & editing, supervision, and funding.

## Competing interests
The authors declare no competing interests.
