## [Transparent Peer review file · Nature Communications]

Neural coding of choice and outcome are modulated by uncertainty in orbitofrontal but not secondary motor cortex

Corresponding Author: Professor Alicia Izquierdo

Version 0:

Reviewer comments:

Reviewer #1

(Remarks to the Author)

The paper by Romero-Sosa et al. titled “Neural coding of choice and outcome are modulated by uncertainty in orbitofrontal but not secondary motor cortex” describes an imaging experiment that sought to determine how neurons in the OFC and M2 region may be recruited during learning under different schedules of reinforcement. Rats were implanted with miniscopes to record GCaMP infected neurons in the OFC or M2 using a GRIN lens in six training sessions. During these training sessions, rats had to learn and reverse deterministic and probabilistic reinforced visual discriminations. Using a SVM to decode neural activity, the authors report that 1) activity in M2 and OFC is encoding choice and outcome and 2) ratio of choice and reward selective neurons increases with uncertainty in OFC but not M2. There are many strengths of the manuscript, including 1) comparison of M2 and lateral OFC, 2) recording during training, 3) inclusion of non-tethered controls, and 4) exciting findings that will be of interest to the readers of Nature communications. There are, however, some significant weakness (outlined below) that make the manuscript feel underdeveloped the claims not well supported.

1. My main concern is that the authors repeatedly argue that the change in OFC encoding across schedules is due to uncertainty and not experience. It is not clear to me however how they can dissociate these things when their experimental design confounds them.
2. I think a justification or at least an explanation is warranted for the block design used. Why did they only reverse one stimulus set instead of reversing both? Why make the uncertainty increase across training? Why not decrease uncertainty to determine if OFC encoding returned to the previous session?
3. The authors use probabilistic reward delivery to manipulate uncertainty. But having within session reversals also introduces uncertainty. I feel like not analyzing the reversal data (or even considering within session block changes as a different form of uncertainty) was a missed opportunity to provide additional support for their argument.
4. I did not understand why several analyses were conducted by the authors, including what the goal of removing cells with the largest and smallest beta coefficients (seems quite obvious that removing strong predictors from analyses would decrease classification accuracy).
5. The results presented in Figure 3 and described in the text were very confusing – I don't understand how the heatmaps in panels A, B, and C relate to panels D, E, and F. For example, it appears that the ratio of side selective cells in OFC neurons during Schedule 1 should be about 0.5 (roughly half of all recorded neurons are showing some evidence of selectivity). But when the ratio is plotted below in D, the value is far below 0.5 (~0.15). If these panels are presenting different measures/metrics, this needs to be clarified then because I could not tell you how or why they are different.
6. One major advantage of miniscopes over fiber photometry is that miniscopes allow one to track the same neuron(s) across time. Did the majority of neurons particularly in the OFC show the uncertainty-induced increase in activity? Because the data being presented in averaged within individual rats, it is hard to get a sense of how robust the effect is within individual neurons.

Minor comments:

1. Figure 3G is not mentioned in the main text

(Remarks on code availability)

I did not see any code available for me to review.

Reviewer #2

(Remarks to the Author)

The authors recorded neural activity in the rat OFC and M2 during a two-arm bandit task where the uncertainty of receiving a reward from each arm was systematically manipulated within and across sessions. Their findings indicate that as uncertainty increased across sessions, OFC activity became better at predicting future choices or outcomes. From this, they argue that the OFC preferentially encodes task variables in uncertain conditions.

The authors perform some impressive technical feats here since getting calcium imaging working in rats, especially from naïve animals, is hard. However, the task design studied here introduces a significant confound due to the fixed order of task conditions. The increase in known uncertainty coincides with a decrease in unknown uncertainty as animals gain more experience with the task. While the authors attempted to account for experience using a GLM, the strong correlation between session number (experience) and uncertainty raises doubts about how effectively the analyses separate these effects. I described this in detail below. Additionally, the manuscript is somewhat descriptive, which makes it challenging to understand the authors' specific claims or the hypotheses tested in each analysis.

In this experiment, two types of uncertainty change in opposite directions across sessions. Known uncertainty increases because the reward probability of the high-reward arm decreases, while that of the low-reward arm increases. In contrast, unknown uncertainty decreases as animals learn the task's deterministic structure, such as the alternation of the high-reward arm every 75 trials and the increase in uncertainty during the third block of each session. As animals become more familiar with the task, they adapt their behavior based on these rules. This is noticeable in animals' performance in figure 1: animals switch their preferred side more quickly at block transitions in later sessions, as shown in Figure 1H, reflecting their ability to anticipate changes. Additionally, animals explore the low-reward arm more frequently during the third block in a session, which aligns with the task design where uncertainty increases in later blocks.

Given this confound, it is difficult to strongly interpret the data and determine whether the improved OFC predictability reflects encoding under conditions of high known uncertainty, or low unknown uncertainty. For example, in Figure 4, the authors showed that certain behavioral measures correlate with decoding accuracy in the OFC. However, changes in these measures could be attributed to either known or unknown uncertainty. For example, perseveration index decreases over sessions. This decline might occur because known uncertainty increases over sessions as the reward probabilities of the two arms become more probabilistic. Alternatively, the decline might reflect a reduction in unknown uncertainty as animals learn with more experience that block changes occur after a specific number of trials.

Although the authors used GLM to control for experience, the tight correlation between uncertainty and experience in this task design limits the strength of conclusions that can be drawn from this data. To address this, the authors could redesign the task so that reward probability changes randomly across blocks and sessions, breaking its correlation with experience. While the authors emphasized the novelty of recording neural data from naïve animals rather than well-trained ones, I believe that to effectively test how representations in M2 and OFC selectively depend on uncertainty, it is important to also manipulate uncertainty levels in well-trained animals, so that there's no confounding between uncertainty and learning effect.

Further, I am a little unsure of how best to interpret the selective increase in OFC decoding with more uncertainty. Even at its best, OFC is weaker than M2 at decoding. So, is M2 not the better region for the rest of the brain to listen to at all times? Given this, the last sentence in the abstract "OFC neurons preferentially encode choices and outcomes under conditions of uncertainty that foster greater reliance on adaptive strategies" is confusing to me.

Minor:

The number of animals is too few. 3 or 4 animals per group seems too few to me. It would be also good to show some raw data in addition to the decoding.

(Remarks on code availability)

Reviewer #3

(Remarks to the Author)

(Remarks on code availability)

Reviewer #4

(Remarks to the Author)

General comments:

By using single-cell calcium imaging with miniscopes in freely behaving rats, Romero-Sosa and colleagues investigated choice and outcome neural selectivity in the orbitofrontal cortex (OFC) and secondary motor cortex (M2) using a restless two-armed bandit task with uncertainty increasing during learning. Remarkably, without extensive training, the rats were able to perform the task and undergo recordings after only a few pretraining sessions, which provides a advantaged behavioral paradigm for studying how different brain regions encode behavioral variables during uncertainty learning in flexible decision-makings.

By decoding calcium neural signals and applying regression models, the authors found that choice decoding accuracy in the M2 remained stable across all certainty conditions, whereas in the OFC it increased with uncertainty. Additionally, single-cell analyses revealed that the proportion of neurons selective for choice and reward increased with uncertainty in OFC but remained unchanged in M2.

Finally, to better examine the increasing involvement of the OFC in encoding behavior variables, the authors combined a direct behavioral measurement (session, latencies), and some decision strategy metrics (win-stay, lose-shift, perseveration, flexibility indexes) into a regression analysis to model the decoding accuracy from two brain areas. The results also showed different patterns between the two brain areas, decoding accuracy for both choice and outcome was significantly correlated with strategy related indexes in OFC, but not in M2.

In general, the authors provide evidence suggesting a different engagement of the OFC and M2 in the adaptive decision with increased uncertainty in a single task. Although the questions and results are interesting, I have several concerns that prevent me from recommending the paper for publication in its current form. If the authors can address all of my concerns then it could be suitable for publication, depending on the outcome of further experiments/analysis.

Major comments:

1. The main concern of the paper is that the choice decoding in M2 is stable over the schedules and OFC is not. The statistic to support this from the decoding analysis is marginally significant, like in table 5 "Area:Schedule" $p=0.0475$. And the decoding of outcome is changing over schedules shown in table 7 "Schedule" $p=0.0494$. This brings up a question of whether this result is reliable. Increasing the N or finding a more sensitive statistical method could address this. I suggest, though it may not be necessary, that the authors consider using predicted probabilities for each trial instead of decoding accuracy for the analyses in Tables 5 and 7, as this might provide additional information for the decoding. Or, the authors could explore different time windows, as the interval between initiation and choice the whole time period corresponds to the animal moving from the central stimulus to the side stimulus.

2. There is an unresolvable correlation between learning and changing uncertainty in the experimental design. Although the authors included session into the regression as a control for learning experience in the behavioral linear model presented in Table 1, if the learning or experience follows a non-linear pattern, it may not be adequately captured by the session variable. The authors tested and recorded animals' behavior over six sessions, with uncertainty progressively increasing across the sessions. However, to better control the experiment, particularly if the goal is to draw strong conclusions about the OFC's involvement in encoding behavioral variables under uncertainty increase, it would be good to include a condition where uncertainty decreases during training for some animals. I understand that starting training with high uncertainty could make it challenging for animals to learn the task. However, the authors could consider reducing uncertainty again after session 6 (reversing the sequence by redoing schedules 2 and 1 after schedules 1, 2, and 3) for additional behavioral testing and recording. If the neural activity from the two brain areas showed a reversed pattern for additional recording sessions, then there is uncertainty increasing and decreasing balanced during learning, it will provide further evidence to support the current conclusions. Or the authors could address this by weakening their claims to acknowledge that the changes in OFC could be due to learning.

3. For the single cell selectivity analysis, I read the methods showed that two separate linear regression models were used to identify neurons as selective by "chosen side", "trial outcome" and also "reward retrieval". If so, then it is not clear how you can clearly identify selectivity by "chosen side" separately from reward expectation, or other rounds in this block design. As the choice is affected by reward expectation, and reward expectation time epoch may still contain choice related information. It's hard to assert that a specific piece of information is exclusively represented within your defined time epoch. Have you tried a linear regression that fits both variables ("chosen side" and "trial outcome") simultaneously?

4. For figure 3F, when counting the reward selective neurons, the authors used "1-s time window centered on head entry into the reward port during rewarded trials versus a corresponding time window from unrewarded trials" (line 190-191). If I understand this clearly, in rewarded trials, an LED cue from the rear reward port signals that a reward is available, whereas in unrewarded trials the LED remains off. If the animal understands the meaning of the cue, it should avoid entering the reward port during unrewarded trials. Then the animal will behave differently for the two trials. So the neural signal from the picked time window is not only modulated by reward, but could also be modulated by the different movements. How to separate reward related signals from the movement related signals? Maybe a small time window around LED on, when animals still have similar movements for both reward and unreward trials is more suitable for this analysis?

5. When explaining the decoding accuracy from two brain areas in figure 2F, the brain areas showed significant main effects (in Table 5), is this main effect related to the two groups of animals showing a difference of choice latency (shown in Table 3)? The difference in choice latency may suggest slightly behavioral differences during the choice epoch. If the authors conduct a video analysis to examine behavioral characteristics, such as movement speed or posture, during the selected time window (figure 2F and G), it could provide valuable evidence to determine whether the main effects of "Area" in Tables 5 and 7 are attributed to differences in behavioral characteristics of the two groups animal or not.

Minor comments:

1. The behavioral related variables for regression models were not defined clearly in the results and methods parts. Like in Table 1, It's not clear if the "TrialNum" is trial number in block, or in session. Is the "session" simply the session number? Is the "block" could be the block number, or block identity? then how to understand the interpretations around line 103-104, behavior "... performance expectedly dipped on block switches when reversals occurred (GLM: $\beta_{\text{block}} = -0.439$, $p < 0.01$)", could the authors define these variables more clearly?
2. For figure 1H, the author showed the behavior for probability of choosing "correct", and for table 1 regression model, it also regressed to "percent correct", I'm a little confused what the "correct" trial meant. I guess it referred to the probability of choosing the better reward option, rather than the trial that the animal could get reward if the animal chose it. Then the $p(\text{correct})$ in figure 1H should use $p(\text{better option})$, and table 1 regresses to "percent better option" instead. The authors should clarify it in the results and methods parts.
3. In this task, the authors used a fixed intertrial interval (ITI, here 10 seconds) and fixed block length (75 trials) for the behavior task. During training, the animals could clearly anticipate when the next trial or block would occur, this design will provide them with a clear expectation of the upcoming stimuli and changes. Could it be possible to use random ITI and block length, and across sessions and animals can make the block length generally equal for different blocks? Then the animals will depend more on their reward experience to adapt their behavior. Or could the authors explain why not use random ITI and block length design?
4. For the time window the authors selected for measuring the decoding accuracy for choice across different schedules (yellow time window in Figure 2F). Based on my understanding, the "initiation" indicated the time animals interacted with the circle presented at the center of the screen. Then the time window between initiation and choice likely represents the period during which the animal moves from the central stimulus to the side stimulus, as there is no indication that the animal is required to maintain fixation on the central circle stimulus. Therefore, neural decoding during this interval likely reflects the choice movement. For the yellow window the author picked, which is quite close to the animal reaching the side stimulus, the regression model showed that in this window the M2 decoding accuracy was not changed across schedules, but OFC increased. But whether this is because decoding accuracy in M2 is high enough to reach a ceiling effect in this picked time window, as if you check the window just after the initiation, seems decoding accuracy from both areas increased across schedules.
5. Line 160-161, "decoder accuracy steadily dropped as more neurons with large beta coefficients were removed", which time window were these beta coefficients regressed on, could the authors explain in more detail in the results and methods parts?
6. For figure 3 A-C, The heatmap plots did not clearly illustrate the trends in the ratio of selective neurons or the changes in coefficients across schedules for the two brain areas. Could this be done in a more proper way, like showing the distribution or histogram of the coefficients from the regression model?
7. For figure 3 G-I, it's not mentioned where these cells come from. Sorting based on peak activity rather than trial number might better highlight how individual cells align with behavioral events?
8. Line 183-187, when the authors tried to explain the possible reason for OFC, "proportion of neurons encoding reward outcome in OFC remain stable despite decrease in the certainty of reward." The authors can simply check the proportion of purely choice selective neurons, purely outcome selective neurons and neurons selective for both. Presenting the ratios across schedules or using pie charts could effectively address the hypothesis?
9. Line 80, it's mentioned "... infuse GCaMP6f ...", could you provide more detailed information about the virus, like the serotype of the virus, and what promoter drives the GCaMP6f expression in the results and methods parts, this will help readers clearly understand the types of neurons being recorded.
10. In the methods parts, the author did not mention what frame per seconds they were used for the data acquisition, if possible, could the author mention more detailed parameters on the miniscope data acquisition?
11. Could the authors include a section in the discussion addressing the limitation of not including any perturbation experiments to confirm the distinct roles of the OFC and M2 in uncertainty learning?
12. There are incorrect table and figure indices in the paper. Such as on line 105 Table 2 not providing the correct information, on line 106-107 Table 5 not providing the correct information. On line 434 Figure 3A-C does not provide the correct information.

(Remarks on code availability)

Version 1:

Reviewer comments:

Reviewer #1

(Remarks to the Author)

I appreciate the revisions the authors made to the manuscript and their responsiveness to the critiques made myself and the other reviews. I particularly appreciate the additional clarification in methodology and analyses the authors did as it improved my understanding of the manuscript. Nevertheless, I have additional comments for the authors regarding the new chemogenetic experiment that I think fell short of what the authors hoped. These are noted below:

- 1) I appreciate the data from the new chemogenetic experiment. Administration of CNO always on the first session (e.g., the most unexpected session) and VEH on the second session (e.g., expected since it is the same as day before) is problematic. The first session is always going to have greater uncertainty than the second (repeat) session so is the effect of CNO just an

order effect? Seems likely since similar patterns exist in the M2 dataset but do not reach statistical significance.

2) Supplementary Figure 7 should replace Figure 5B. Figure 5B raises way more questions (e.g., what is the DV on the y axis even represent? How did the p(correct) change across the session? How do M2 effects compared to OFC effects that the authors report? What about ascending and descending uncertainty?). All these questions can be answered from supplement Fig 7 and not in Figure 5B, which raised more questions to me than answers.

3) The chemogenetic experiment does attempt to address the larger concern noted by reviewers that uncertainty in reinforcement schedules co-varies with experience. However, I am not completely convinced with this additional and suggest that the authors clear state that these two distinct forms of uncertainty exist in the current study as a limitation in the discussion.

(Remarks on code availability)

Unfortunately I did not have time to review the code.

Reviewer #2

(Remarks to the Author)

We appreciate the authors' efforts in conducting an additional experiment to break the inherent correlation between uncertainty and experience present in the original experimental design and address the causal role of OFC in decision making under conditions of high uncertainty. We agree that the new experimental design effectively disentangles uncertainty from experience. However, we find that the evidence from presented results could be stronger with improved statistical analysis.

From our understanding, the authors calculated performance within a 15-trial moving window and treated each of these windows as independent samples. Using a moving average is concerning, as it creates repeated usage of the same data points. Each individual trial contributes to multiple (in this case, up to 15) overlapping windows. This can artificially inflate the sample size and generate high correlation across samples, thereby violating statistical assumptions of independence and potentially biasing statistical results.

We suggest that using non-overlapping window would be more statistically appropriate. We appreciate that the GLM used here accounts for mixed effects and that is great. Nevertheless, given the complexity of the nested experimental design (windows nested within blocks, sessions, schedules, and rats), applying a hierarchical bootstrap method (<https://pmc.ncbi.nlm.nih.gov/articles/PMC7906290/>) could provide a more rigorous statistical test of significance.

Alternatively, a more conservative approach would be to calculate the performance for each animal within each session and directly test whether the drug impacts this measure (each animal will be treated as a different independent sample in this case).

(Remarks on code availability)

The code looks okay to me.

Reviewer #3

(Remarks to the Author)

(Remarks on code availability)

Reviewer #4

(Remarks to the Author)

The authors have carefully considered my concerns and provided detailed and satisfying responses to most of the comments I raised. In particular, they provided new data with decent numbers of animals, this new experiment not only provides a causal test of the distinct contributions of M2 and OFC in this task, also provides a well controlled paradigm to dissociate uncertainty across different schedules and learning.

However, my 3rd major comment has not been fully addressed. In this comment, I raised concerns about the use of separate regression models may not be the best way to identify neurons as selective for "chosen side," "trial outcome," and "reward retrieval." While I appreciate the authors' effort to mitigate this issue by selecting narrow time windows, and also conducted an additional regression analysis, but these approaches do not entirely resolve the concern. From Figure 3A, it appears that there's still lots of choice-related activities does not return to around 0 within the time window selected by the authors (+1s relative to choice). This raises ambiguity about whether the narrow time windows the author selected for calculating single cell selectivity is still overlap across these different variables, particularly in the block design task. If choice-related information persists, it may be mistakenly attributed to outcome or reward selectivity in this case. To more clearly dissociate the selectivity, fit both "chosen side" and "trial outcome" within a single regression model, or at least consider including

“chosen side” as random effect when calculating selectivity for “trial outcome”/“reward retrieval”. Or the authors could show that almost all choice-selective neurons show no choice selectivity during the “trial outcome” or “reward retrieval” time windows.

(Remarks on code availability)

Version 2:

Reviewer comments:

Reviewer #1

(Remarks to the Author)

I appreciate the authors responding to my additional concerns. However, I am not convinced that what they are observing in the chemogenetic experiment is not just an order effect. This needs to be addressed or at the very least acknowledged in the manuscript as it is a significant limitation in the interpretation of the results.

(Remarks on code availability)

Reviewer #2

(Remarks to the Author)

Despite the authors' revised analysis, our original concern remains unresolved. In the previous revision, authors used a 15-trial sliding window to calculate performance $P(\text{better})$. This method reuses each trial multiple times, leading to non-independent samples. Because these performance estimates were then used in statistical tests, this repeated use of data violates the assumption of sample independence in statistical tests.

In the rebuttal, the authors claimed to have addressed this issue by using a non-overlapping window. However, according to the methods section, the revised method appears to compute $P(\text{better})$ at each trial by averaging all choices from the first trial up to the current one. This essentially means that the first trial contributes to the estimates for trials 1 through n . This is not a non-overlapping window. It is a maximally overlapping one. Therefore, the revised method still results in non-independent samples, and the statistical concern remains unresolved.

(Remarks on code availability)

Reviewer #3

(Remarks to the Author)

(Remarks on code availability)

Reviewer #4

(Remarks to the Author)

The authors have carefully addressed all of my concerns, and the main results are still good enough to support the different roles of M2 and OFC under different uncertainty conditions in their task. I support publication of this study.

Minor comment:

Typo line 317 "We als find..." should be "We also find"

(Remarks on code availability)

Version 3:

Reviewer comments:

Reviewer #2

(Remarks to the Author)

I appreciate the authors addressing the concerns I raised. The authors now fit a glm to predict choice (1=better option, 0=worse option), and then use that predicted output from glm (p_{better}) for their significance test of modulation effect. So

instead of using the raw choice that animals made to test significance of modulation, they use inferred p better for the significance test. This is fine but the authors should consider mentioning this important analytical choice and its effect on the interpretation of their results.

(Remarks on code availability)

Reviewer #3

(Remarks to the Author)

(Remarks on code availability)

REVIEWER COMMENTS

We have provided code for peer review: <https://codeocean.com/capsule/0863492/tree>

Reviewer #1

The paper by Romero-Sosa et al. titled “Neural coding of choice and outcome are modulated by uncertainty in orbitofrontal but not secondary motor cortex” describes an imaging experiment that sought to determine how neurons in the OFC and M2 region may be recruited during learning under different schedules of reinforcement. Rats were implanted with miniscopes to record GCaMP infected neurons in the OFC or M2 using a GRIN lens in six training sessions. During these training sessions, rats had to learn and reverse deterministic and probabilistic reinforced visual discriminations. Using a SVM to decode neural activity, the authors report that 1) activity in M2 and OFC is encoding choice and outcome and 2) ratio of choice and reward selective neurons increases with uncertainty in OFC but not M2. There are many strengths of the manuscript, including 1) comparison of M2 and lateral OFC, 2) recording during training, 3) inclusion of non-tethered controls, and 4) exciting findings that will be of interest to the readers of Nature communications. There are, however, some significant weakness (outlined below) that make the manuscript feel underdeveloped the claims not well supported.

We thank the reviewer for this positive feedback.

1. My main concern is that the authors repeatedly argue that the change in OFC encoding across schedules is due to uncertainty and not experience. It is not clear to me however how they can dissociate these things when their experimental design confounds them.

We thank the reviewer for pointing out the overlap of uncertainty with experience in our experimental design. In the original manuscript, we dealt with this statistically by including session as a covariate. To further dissociate contributions of experience versus uncertainty, we now include new data from n=18 animals in which DREADDs were used to inactivate either M2 or OFC during learning under uncertainty. In this additional experiment, rats were introduced to different schedules first in order of increasing uncertainty and then in reverse order of decreasing uncertainty. Each schedule was administered across two days. We found that there was no effect of session (i.e., experience), so we collapsed schedules across learning, finding that the most significant involvement of OFC on performance accuracy was in the most uncertain schedule, and the most significant involvement of M2 was in the most certain one. We find similar results in WinStay strategies- supporting our findings in Figure 4. These results are included as new data, new analyses, and a new **Figure 5**. We have also added a **Supplementary Figure 7** where we show a histogram for every session separated by whether they were part of the ascending or descending order. These results are concordant with the calcium imaging results.

2. I think a justification or at least an explanation is warranted for the block design used. Why did they only reverse one stimulus set instead of reversing both? Why make the uncertainty increase across training? Why not decrease uncertainty to determine if OFC encoding returned to the previous session?

We now include a few sentences explanation for the block reversal design in the **Methods**. A main reason for the block reversal design was to keep the animals engaged and prevent them from falling into a motor habit of always responding to one side. The reviewer is correct that we did not decrease the uncertainty and we have addressed this in a new **Figure 5**. These data are from an additional n=18 rats with DREADDs in OFC (n=9) or M2 (n=9). Because rats often

decrease engagement in the task if we start them on an uncertain schedule, it was necessary to first increase and then decrease uncertainty. Additionally, we have performed further block analysis included in **Supplementary Figure 5**.

3. The authors use probabilistic reward delivery to manipulate uncertainty. But having within session reversals also introduces uncertainty. I feel like not analyzing the reversal data (or even considering within session block changes as a different form of uncertainty) was a missed opportunity to provide additional support for their argument.

We appreciate this is a missed opportunity so we have now included reversal/block analyses as a **Supplementary Figure 5**. Block analyses show there is a larger ratio of block-selective neurons in M2 compared to OFC throughout all of the three different epochs. We think this is evidence that M2 activity represents more of the trial structure whereas OFC neuronal activity represents trial outcome and updating the value of each choice. This has been added to the Discussion.

4. I did not understand why several analyses were conducted by the authors, including what the goal of removing cells with the largest and smallest beta coefficients (seems quite obvious that removing strong predictors from analyses would decrease classification accuracy).

We regret not clearly communicating the rationale for our analyses- we add a justification to the manuscript. The main purpose of re-running the decoders and removing the neurons with the largest beta coefficients first was to co-validate results of the population level analysis (i.e., the decoders) with the single cell analysis (i.e., the GLMs). This is why we include this as a **Supplementary Figure 5** as opposed to a main figure.

5. The results presented in Figure 3 and described in the text were very confusing – I don't understand how the heatmaps in panels A, B, and C relate to panels D, E, and F. For example, it appears that the ratio of side selective cells in OFC neurons during Schedule 1 should be about 0.5 (roughly half of all recorded neurons are showing some evidence of selectivity). But when the ratio is plotted below in D, the value is far below 0.5 (~0.15). If these panels are presenting different measures/metrics, this needs to be clarified then because I could not tell you how or why they are different.

We have clarified the descriptions of these panels in the figure caption. Briefly, all neurons shown in panels A-C are side selective, with heatmap colors indicating the ratio of left/right or reward/unrewarded- selective neurons. For example, panel A shows that in M2 there is a larger proportion of right than left preferring side selective cells in schedule 1, and by schedule 3 the ratio actually reverses and so there are more neurons that exhibit greater activity during the left side choice.

6. One major advantage of miniscopes over fiber photometry is that miniscopes allow one to track the same neuron(s) across time. Did the majority of neurons particularly in the OFC show the uncertainty-induced increase in activity? Because the data being presented in averaged within individual rats, it is hard to get a sense of how robust the effect is within individual neurons.

We thank the reviewer for suggesting this. We have now conducted cell registration- testing one session (session 2) using training data from the activity of the same neurons from the previous session (session 1), within the same schedule. The results, presented in **Supplementary Figure 6**, are largely consistent with our main **Figure 2**. Interestingly, cross-registered neurons are more useful in decoding Chosen Side than Trial Outcome. This suggests the same neurons consistently encode the former across days, but different cells may encode the latter from one day to the next even within the same schedule.

Minor comments:

1. Figure 3G is not mentioned in the main text
Thank you, we now mention it in the main text.

Reviewer #1 (Remarks on code availability):

I did not see any code available for me to review.
We have uploaded the revised code for review on Cloud.

Reviewer #2

The authors recorded neural activity in the rat OFC and M2 during a two-arm bandit task where the uncertainty of receiving a reward from each arm was systematically manipulated within and across sessions. Their findings indicate that as uncertainty increased across sessions, OFC activity became better at predicting future choices or outcomes. From this, they argue that the OFC preferentially encodes task variables in uncertain conditions.

The authors perform some impressive technical feats here since getting calcium imaging working in rats, especially from naïve animals, is hard. However, the task design studied here introduces a significant confound due to the fixed order of task conditions. The increase in known uncertainty coincides with a decrease in unknown uncertainty as animals gain more experience with the task. While the authors attempted to account for experience using a GLM, the strong correlation between session number (experience) and uncertainty raises doubts about how effectively the analyses separate these effects. I described this in detail below. Additionally, the manuscript is somewhat descriptive, which makes it challenging to understand the authors' specific claims or the hypotheses tested in each analysis.

We thank the reviewer for their incisive comments. We now better define what we mean by uncertainty and add an experiment to address the confounding of experience and uncertainty. More details on these changes are provided below.

In this experiment, two types of uncertainty change in opposite directions across sessions. Known uncertainty increases because the reward probability of the high-reward arm decreases, while that of the low-reward arm increases. In contrast, unknown uncertainty decreases as animals learn the task's deterministic structure, such as the alternation of the high-reward arm every 75 trials and the increase in uncertainty during the third block of each session. As animals become more familiar with the task, they adapt their behavior based on these rules. This is noticeable in animals' performance in figure 1: animals switch their preferred side more quickly at block transitions in later sessions, as shown in Figure 1H, reflecting their ability to anticipate changes. Additionally, animals explore the low-reward arm more frequently during the third block in a session, which aligns with the task design where uncertainty increases in later blocks. The reviewer provides a thorough explanation and critique of the overlap of experience and uncertainty. What we state here as "uncertainty" we mean as stochasticity (i.e., moment-to-moment probabilistic outcomes), but not volatility (i.e., rate or pattern of reversals), which as the reviewer already notes, does not change over time; the rats gain expertise with this. We agree that to disentangle this would require a complete redesign of our experiments (a 2 x 2 design) that manipulates both. We are more interested in the stochasticity so we now clarify this in the Introduction. We also acknowledge it is insufficient to only deal with experience (volatility, or as the reviewer refers to it, 'known uncertainty') statistically as "session." To more fully address this concern, we now include new data following DREADDs inactivation of these regions to causally test their influence in 'ascending' and 'descending' uncertainty (i.e., stochasticity) that controls

for experience with volatility. We think that the learning curves from this additional cohort of $n=18$ animals clearly show that rats do not “anticipate” the reversals since the drop in $p(\text{Better})$ only occurs after the reversal / block switch. In this larger cohort of animals we see a more consistent adaptation to the reversal across all schedules:

Given this confound, it is difficult to strongly interpret the data and determine whether the improved OFC predictability reflects encoding under conditions of high known uncertainty, or low unknown uncertainty. For example, in Figure 4, the authors showed that certain behavioral measures correlate with decoding accuracy in the OFC. However, changes in these measures could be attributed to either known or unknown uncertainty. For example, perseveration index decreases over sessions. This decline might occur because known uncertainty increases over sessions as the reward probabilities of the two arms become more probabilistic. Alternatively, the decline might reflect a reduction in unknown uncertainty as animals learn with more experience that block changes occur after a specific number of trials.

We thank the Reviewer for pointing out this issue, which we think we now fully address with our

causal experiment (Figure 5). Now that we uncover involvement of strategy (WinStay) in OFC across more schedules than in M2, specifically in our chemogenetic experiment that controls for experience, we are more confident in our ability to strongly link OFC activity to adaptive strategies under uncertainty.

Although the authors used GLM to control for experience, the tight correlation between uncertainty and experience in this task design limits the strength of conclusions that can be drawn from this data. To address this, the authors could redesign the task so that reward probability changes randomly across blocks and sessions, breaking its correlation with experience. While the authors emphasized the novelty of recording neural data from naïve animals rather than well-trained ones, I believe that to effectively test how representations in M2 and OFC selectively depend on uncertainty, it is important to also manipulate uncertainty levels in well-trained animals, so that there’s no confounding between uncertainty and learning effect.

We agree. In the additional DREADDs experiment, animals had the same experience and uncertainty (i.e., stochasticity) increased and then decreased. We found a more significant involvement of OFC in choosing the better option on the more uncertain schedules (whether ascending or descending), and a more significant involvement of M2 with the more certain schedule (again, whether it was ascending or descending). With $n=18$ additional animals we now include new data following inactivation of OFC vs. M2 to causally test the influence of these regions in ‘ascending’ and ‘descending’ uncertainty. This is included as a new Figure 5.

Further, I am a little unsure of how best to interpret the selective increase in OFC decoding with more uncertainty. Even at its best, OFC is weaker than M2 at decoding. So, is M2 not the better

region for the rest of the brain to listen to at all times? Given this, the last sentence in the abstract “OFC neurons preferentially encode choices and outcomes under conditions of uncertainty that foster greater reliance on adaptive strategies” is confusing to me.

Thank you for pointing out this lack of clarity. While it is true that M2 is clearly “the better region for the rest of the brain to listen to” (as the reviewer states)- we say as much in the Discussion when discussing Brain-Machine interface applications, and we do think that OFC activity is much more linked to adaptive strategy – as observed both in the imaging dataset and also in the results of the chemogenetic experiment. We add more on this to the Discussion.

Minor:

The number of animals is too few. 3 or 4 animals per group seems too few to me. It would be also good to show some raw data in addition to the decoding.

We have added n=18 animals in a DREADDs experiment. Additionally, in Figure 3 G, H, and I we show examples of raw data from selective neurons.

Reviewer #3

I co-reviewed this manuscript with one of the reviewers who provided the listed reports. This is part of the NatureCommunications initiative to facilitate training in peer review and to provide appropriate recognition for Early Career Researchers who co-review manuscripts.

We thank this reviewer for co-reviewing.

Reviewer #4

General comments:

By using single-cell calcium imaging with miniscopes in freely behaving rats, Romero-Sosa and colleagues investigated choice and outcome neural selectivity in the orbitofrontal cortex (OFC) and secondary motor cortex (M2) using a restless two-armed bandit task with uncertainty increasing during learning. Remarkably, without extensive training, the rats were able to perform the task and undergo recordings after only a few pretraining sessions, which provides a advantaged behavioral paradigm for studying how different brain regions encode behavioral variables during uncertainty learning in flexible decision-makings.

By decoding calcium neural signals and applying regression models, the authors found that choice decoding accuracy in the M2 remained stable across all certainty conditions, whereas in the OFC it increased with uncertainty. Additionally, single-cell analyses revealed that the proportion of neurons selective for choice and reward increased with uncertainty in OFC but remained unchanged in M2.

Finally, to better examine the increasing involvement of the OFC in encoding behavior variables, the authors combined a direct behavioral measurement (session, latencies), and some decision strategy metrics (win-stay, lose-shift, perseveration, flexibility indexes) into a regression analysis to model the decoding accuracy from two brain areas. The results also showed different patterns between the two brain areas, decoding accuracy for both choice and outcome was significantly correlated with strategy related indexes in OFC, but not in M2.

In general, the authors provide evidence suggesting a different engagement of the OFC and M2 in the adaptive decision with increased uncertainty in a single task. Although the questions and results are interesting, I have several concerns that prevent me from recommending the paper for publication in its current form. If the authors can address all of my concerns then it could be suitable for publication, depending on the outcome of further experiments/analysis.

Major comments:

1. The main concern of the paper is that the choice decoding in M2 is stable over the schedules and OFC is not. The statistic to support this from the decoding analysis is marginally significant, like in table 5 “Area:Schedule” $p=0.0475$. And the decoding of outcome is changing over schedules shown in table 7 “Schedule” $p=0.0494$. This brings up a question of whether this result is reliable. Increasing the N or finding a more sensitive statistical method could address this. I suggest, though it may not be necessary, that the authors consider using predicted probabilities for each trial instead of decoding accuracy for the analyses in Tables 5 and 7, as this might provide additional information for the decoding. Or, the authors could explore different time windows, as the interval between initiation and choice the whole time period corresponds to the animal moving from the central stimulus to the side stimulus.

We thank this Reviewer for these suggestions. We do agree that we need to show additional evidence of our main contention that OFC and M2 have complementary roles in uncertainty and certainty, respectively, by adding more animals. We have now added a causal experiment with **n=18 additional animals**. More details are provided below.

2. There is an unresolvable correlation between learning and changing uncertainty in the experimental design. Although the authors included session into the regression as a control for learning experience in the behavioral linear model presented in Table 1, if the learning or experience follows a non-linear pattern, it may not be adequately captured by the session variable. The authors tested and recorded animals’ behavior over six sessions, with uncertainty progressively increasing across the sessions. However, to better control the experiment, particularly if the goal is to draw strong conclusions about the OFC’s involvement in encoding behavioral variables under uncertainty increase, it would be good to include a condition where uncertainty decreases during training for some animals. I understand that starting training with high uncertainty could make it challenging for animals to learn the task. However, the authors could consider reducing uncertainty again after session 6 (reversing the sequence by redoing schedules 2 and 1 after schedules 1, 2, and 3) for additional behavioral testing and recording. If the neural activity from the two brain areas showed a reversed pattern for additional recording sessions, then there is uncertainty increasing and decreasing balanced during learning, it will provide further evidence to support the current conclusions. Or the authors could address this by weakening their claims to acknowledge that the changes in OFC could be due to learning.

We thank the reviewer for raising this issue. To address this concern, we now include new data following DREADDs inactivation of these regions to causally test the influence of these regions. In this additional experiment, animals had the same experience but uncertainty increased and then decreased. We found no difference in the ascending or descending phases of the same schedule (i.e., Schedule 1 ascending vs Schedule 1 descending), and we find a more significant involvement of OFC with the more uncertain schedule, and a more significant involvement of M2 with the more certain schedule. We find a similar pattern for WinStay strategies. This is included as a new **Figure 5**.

3. For the single cell selectivity analysis, I read the methods showed that two separate linear regression models were used to identify neurons as selective by “chosen side”, “trial outcome” and also “reward retrieval”. If so, then it is not clear how you can clearly identify selectivity by “chosen side” separately from reward expectation, or other rounds in this block design. As the choice is affected by reward expectation, and reward expectation time epoch may still contain choice related information. It’s hard to assert that a specific piece of information is exclusively represented within your defined time epoch. Have you tried a linear regression that fits both variables (“chosen side” and “trial outcome”) simultaneously?

The reviewer points out the limitations of a naturalistic, freely-moving design. Indeed, we selected the narrow time windows to limit and constrain what the animal’s movement and

behavior could be. Though we have not conducted a regression model with both “chosen side” and “trial outcome” simultaneously, we have looked deeper into individual neuron selectivity by conducting an analysis where block is one of the regressors with “chosen side” and “trial outcome” as covariates. The results reveal that there are consistently more “Block Selective” neurons in M2 than OFC. This was the case for all three time windows investigated (before choice, before trial outcome, and during reward retrieval). These results are shown in **Supplementary Figure 5**.

4. For figure 3F, when counting the reward selective neurons, the authors used “1-s time window centered on head entry into the reward port during rewarded trials versus a corresponding time window from unrewarded trials” (line 190-191). If I understand this clearly, in rewarded trials, an LED cue from the rear reward port signals that a reward is available, whereas in unrewarded trials the LED remains off. If the animal understands the meaning of the cue, it should avoid entering the reward port during unrewarded trials. Then the animal will behave differently for the two trials. So the neural signal from the picked time window is not only modulated by reward, but could also be modulated by the different movements. How to separate reward related signals from the movement related signals? Maybe a small time window around LED on, when animals still have similar movements for both reward and unreward trials is more suitable for this analysis?

The ideal condition for this comparison would be to have the rat check the reward port before the reward cue, however, this would require a complete redesign of the experiment. By looking at a time point that is approximately the median latency of reward collection after choice, we were hoping to approximate this timing. If we focused our analysis when the LED is still on, as suggested, it would overlap with the “Reward Cue” epoch that occurs right before the cue. With the slow temporal dynamics of GCamp6f we were worried about not capturing different dynamics, which is why we chose these windows.

5. When explaining the decoding accuracy from two brain areas in figure 2F, the brain areas showed significant main effects (in Table 5), is this main effect related to the two groups of animals showing a difference of choice latency (shown in Table 3)? The difference in choice latency may suggest slightly behavioral differences during the choice epoch. If the authors conduct a video analysis to examine behavioral characteristics, such as movement speed or posture, during the selected time window (figure 2F and G), it could provide valuable evidence to determine whether the main effects of “Area” in Tables 5 and 7 are attributed to differences in behavioral characteristics of the two groups animal or not.

We thank the reviewer for pointing out the possibility that a difference in choice latency might explain the decoder accuracy results. We believe there is strong evidence against this possibility: in Table 14 we include all of the latencies as regressors as well as their interaction with Area to investigate if any of these variables influence decoder accuracy for Chosen Side. Choice latency was not a significant predictor of decoder accuracy, nor did it interact with Area.

Minor comments:

1. The behavioral related variables for regression models were not defined clearly in the results and methods parts. Like in Table 1, It’s not clear if the “TrialNum” is trial number in block, or in session. Is the “session” simply the session number? Is the “block” could be the block number, or block identity? then how to understand the interpretations around line 103-104, behavior “... performance expectedly dipped on block switches when reversals occurred (GLM: $\beta_{\text{block}} = -0.439, p < 0.01$)”, could the authors define these variables more clearly?

We regret not having been clear. We have added more detail in the Methods (Data Analysis section) to clearly define these predictors.

2. For figure 1H, the author showed the behavior for probability of choosing “correct”, and for table 1 regression model, it also regressed to “percent correct”, I’m a little confused what the “correct” trial meant. I guess it referred to the probability of choosing the better reward option, rather than the trial that the animal could get reward if the animal chose it. Then the p(correct) in figure 1H should use p(better option), and table 1 regresses to “percent better option” instead. The authors should clarify it in the results and methods parts.

We have changed the labels in Figure 1E, Supplementary Figure 2, Table 1, and Supplementary Table 1 to **p(Better)**.

3. In this task, the authors used a fixed intertrial interval (ITI, here 10 seconds) and fixed block length (75 trials) for the behavior task. During training, the animals could clearly anticipate when the next trial or block would occur, this design will provide them with a clear expectation of the upcoming stimuli and changes. Could it be possible to use random ITI and block length, and across sessions and animals can make the block length generally equal for different blocks? Then the animals will depend more on their reward experience to adapt their behavior. Or could the authors explain why not use random ITI and block length design?

We thank the reviewer for pointing out this concern about the behavioral task design. Here, the fully predictive block switch contributes to what Reviewer 2 calls a “known uncertainty” and it ultimately helps us make the case that animals gain expertise in this aspect of the task because it never changes, whereas the stochasticity (what we are really interested in) changes. While it’s true that having a block switch every 75 trials could lead to a “prediction” of a switch, we do not see convincing evidence of this in the data. In **Figure 1H** we show an average sliding window that includes only past trials and there is no anticipatory drop in accuracy in choosing the correct option *before* the reversal. Thus, there is no strong evidence there is strong prediction of the block switch.

We do agree that for future experiments we should make the block switch more unpredictable: for example, implementing a variable ITI with jitter, pulled from a normal distribution centered around 5 seconds. We thank the reviewer for the suggestion.

4. For the time window the authors selected for measuring the decoding accuracy for choice across different schedules (yellow time window in Figure 2F). Based on my understanding, the “initiation” indicated the time animals interacted with the circle presented at the center of the screen. Then the time window between initiation and choice likely represents the period during which the animal moves from the central stimulus to the side stimulus, as there is no indication that the animal is required to maintain fixation on the central circle stimulus. Therefore, neural decoding during this interval likely reflects the choice movement. For the yellow window the author picked, which is quite close to the animal reaching the side stimulus, the regression model showed that in this window the M2 decoding accuracy was not changed across schedules, but OFC increased. But whether this is because decoding accuracy in M2 is high enough to reach a ceiling effect in this picked time window, as if you check the window just after the initiation, seems decoding accuracy from both areas increased across schedules.

We thank the reviewer for suggesting to investigate the time window right after Initiation. We do not find any significant differences in decoder accuracy either across schedules or between different brain areas during this time interval (see Table below). Decoder accuracy is above chance which means that both M2 and OFC contain a good amount of information regarding what the future choice will be at that point.

5. Line 160-161, “decoder accuracy steadily dropped as more neurons with large beta coefficients were removed”, which time window were these beta coefficients regressed on, could the authors explain in more detail in the results and methods parts?

We have added more description of the time windows in the Method section *Single Cell Selectivity GLM*.

Comparing Chosen Side Decoder Accuracy Across all Schedules in M2 and OFC							
$\gamma = \text{Decoder Accuracy after Initiation}$							
Formula	$\gamma \sim [1 + \text{Area} * \text{Schedule} + \text{Session} + (1 \text{RatID} : \text{Session})]$						
Coefficients	β	SE	tStat	DF	P	CIL	CIU
(Intercept)	0.7151	0.1063	6.7262	37	<0.0001	0.4997	0.9306
Session	-0.0930	0.0641	-1.4510	37	0.1552	-0.2230	0.0369
Schedule	0.0311	0.0240	1.2953	37	0.2032	-0.0175	0.0797
Area	-0.0104	0.0685	-0.1523	37	0.8798	-0.1492	0.1284
Area:Schedule	0.0035	0.0297	0.1188	37	0.9061	-0.0566	0.0637

6. For figure 3 A-C, The heatmap plots did not clearly illustrate the trends in the ratio of selective neurons or the changes in coefficients across schedules for the two brain areas. Could this be done in a more proper way, like showing the distribution or histogram of the coefficients from the regression model?

Our main goal with panels A-C in Figure 3 was to showcase whether the significant neurons were firing more for Left/Right or Rewarded/Unrewarded- and it shows the ratio of selective to non-selective neurons changes with schedules. In contrast, the distribution of coefficients does not change much across schedules- they are normally distributed centered around 0.

7. For figure 3 G-I, it's not mentioned where these cells come from. Sorting based on peak activity rather than trial number might better highlight how individual cells align with behavioral events?

We have changed the figure caption to note which region each of the example neurons was imaged in. As for using a peak sorted plot: typically, those plots are most useful to show how multiple neurons become active in relation to the epoch of interest. However, here we have multiple trials and therefore need to show the activity throughout multiple blocks within a single session. Impressively, these example neurons are reliably active despite not being peak sorted.

8. Line 183-187, when the authors tried to explain the possible reason for OFC, “proportion of neurons encoding reward outcome in OFC remain stable despite decrease in the certainty of reward.” The authors can simply check the proportion of purely choice selective neurons, purely outcome selective neurons and neurons selective for both. Presenting the ratios across schedules or using pie charts could effectively address the hypothesis?

We have now checked to see if there is a significant difference between the ratio of neurons that is purely side-selective, is purely reward-selective, or both and have found only a significantly greater number of neurons selective for both in M2 (see tables below). This result confirms what we see in Figure 3D-F that single cell selectivity is generally stronger for all task features in M2. Thus, we think our analysis as presented has finer-grained resolution for OFC.

9. Line 80, it's mentioned "... infuse GCaMP6f ...", could you provide more detailed information

Comparing Ratio of both Side & Reward Selective out of all selective Cells Across all Schedules in M2 and OFC							
$\gamma = \text{Ratio of Side \& Reward Selective Cells}$							
Formula	$\gamma \sim [1 + \text{Area} * \text{Schedule} + \text{Session} + (1 \text{RatID} : \text{Session})]$						
Coefficients	β	SE	tStat	DF	P	CIL	CIU
(Intercept)	0.3613	0.0909	3.9749	37	0.0003	0.1771	0.5455
Session	-0.0220	0.0205	-1.0708	37	0.2912	-0.0635	0.0196
Schedule	-0.0314	0.0586	-0.5356	37	0.5954	-0.1500	0.0873
Area	-0.1390	0.0548	-2.5363	37	0.0156	-0.2501	-0.0280
Area:Schedule	0.0447	0.0254	1.7624	37	0.0863	-0.0067	0.0961

Comparing Ratio of Purely Side Selective out of all Selective Cells Across all Schedules in M2 and OFC							
$\gamma = \text{Ratio of Side Selective Cells}$							
Formula	$\gamma \sim [1 + \text{Area} * \text{Schedule} + \text{Session} + (1 \text{RatID} : \text{Session})]$						
Coefficients	β	SE	tStat	DF	P	CIL	CIU
(Intercept)	0.3757	0.1370	2.7431	37	0.0093	0.0982	0.6532
Session	0.0376	0.0309	1.2167	37	0.2314	-0.0250	0.1002
Schedule	0.0323	0.0882	0.3657	37	0.7167	-0.1465	0.2111
Area	0.0081	0.0826	0.0985	37	0.9221	-0.1592	0.1755
Area:Schedule	-0.0310	0.0382	-0.8099	37	0.4232	-0.1084	0.0465

Comparing Ratio of Purely Reward Selective out of all Selective Cells Across all Schedules in M2 and OFC							
$\gamma = \text{Ratio of Reward Selective Cells}$							
Formula	$\gamma \sim [1 + \text{Area} * \text{Schedule} + \text{Session} + (1 \text{RatID} : \text{Session})]$						
Coefficients	β	SE	tStat	DF	P	CIL	CIU
(Intercept)	0.2630	0.1248	2.1066	37	0.0420	0.0100	0.5159
Session	-0.0156	0.0282	-0.5552	37	0.5821	-0.0727	0.0414
Schedule	-0.0009	0.0804	-0.0112	37	0.9911	-0.1639	0.1621
Area	0.1309	0.0753	1.7387	37	0.0904	-0.0216	0.2835
Area:Schedule	-0.0138	0.0349	-0.3948	37	0.6952	-0.0844	0.0569

about the virus, like the serotype of the virus, and what promoter drives the GCaMP6f expression in the results and methods parts, this will help readers clearly understand the types of neurons being recorded.

Thank you for alerting us to this omission. The information has been added to the methods.

10. In the methods parts, the author did not mention what frame per seconds they were used for the data acquisition, if possible, could the author mention more detailed parameters on the miniscope data acquisition?

All imaging was conducted at a rate of 20 frames-per-second (fps). This detail has been added to the Methods.

11. Could the authors include a section in the discussion addressing the limitation of not including any perturbation experiments to confirm the distinct roles of the OFC and M2 in uncertainty learning?

Chemogenetic perturbation experiments with n=18 have now been added, as described above and are highlighted in **Figure 5**.

12. There are incorrect table and figure indices in the paper. Such as on line 105 Table 2 not providing the correct information, on line 106-107 Table 5 not providing the correct information. On line 434 Figure 3A-C does not provide the correct information.

We regret these errors and thank the reviewer for pointing them out. They have been fixed.

REVIEWER COMMENTS

Reviewer #1 (Remarks to the Author):

I appreciate the revisions the authors made to the manuscript and their responsiveness to the critiques made myself and the other reviews. I particularly appreciate the additional clarification in methodology and analyses the authors did as it improved my understanding of the manuscript. Nevertheless, I have additional comments for the authors regarding the new chemogenetic experiment that I think fell short of what the authors hoped. These are noted below:

We thank this reviewer for their feedback.

1) I appreciate the data from the new chemogenetic experiment. Administration of CNO always on the first session (e.g., the most unexpected session) and VEH on the second session (e.g., expected since it is the same as day before) is problematic. The first session is always going to have greater uncertainty than the second (repeat) session so is the effect of CNO just an order effect? Seems likely since similar patterns exist in the M2 dataset but do not reach statistical significance.

We have a recent publication showing that the effect of chemogenetic inhibition of OFC is sensitive to order in that it is more involved in first reversals (Aguirre et al. J Neurosci 2024). Others have also shown this to be the case (Groman et al. 2019; Verharen et al. 2020), so we made the decision to always give CNO on the first sessions. While it is true that drug was always presented in CNO-VEH in Session 1-2 of each schedule, we were mainly interested in the pattern across schedules, not within a schedule, and across M2 vs. OFC. Thus, even with a drug order effect, we can still compare the role of OFC and M2 inhibition across schedules. We also clarify that we in fact did not see the same effect in M2 as the reviewer stated, which we take to mean even more convincingly that the two regions are supporting different facets of learning. We have clarified this in the manuscript.

2) Supplementary Figure 7 should replace Figure 5B. Figure 5B raises way more questions (e.g., what is the DV on the y axis even represent? How did the p(correct) change across the session? How do M2 effects compared to OFC effects that the authors report? What about ascending and descending uncertainty?). All these questions can be answered from supplement Fig 7 and not in Figure 5B, which raised more questions to me than answers.

Thank you for this suggestion. We have replaced Figure 5B with the schedule-wise comparison in Supplementary Figure 7, and provide supporting statistics. On Reviewer 2's suggestion to calculate pCorr without sliding windows (see below), we have re-run the statistics, finding significant interactions of schedule order x drug, and can then do postdocs. Results are largely similar to before, with OFC but not M2 inhibition producing the largest attenuating effect on (most uncertain) Schedule 3 accuracy. Conversely, M2 (but not OFC) inhibition produces the largest attenuating effect on (most certain) Schedule 1.

3) The chemogenetic experiment does attempt to address the larger concern noted by reviewers that uncertainty in reinforcement schedules co-varies with experience. However, I am not completely convinced with this additional and suggest that the authors clear state that these two distinct forms of uncertainty exist in the current study as a limitation in the discussion.

With the additional analyses recommended by Reviewer 2 to calculate pCorr without sliding windows, we are more convinced of the involvement of OFC in learning under uncertainty, not experience. This was the primary reason we implemented the ascending and descending design. Indeed, we observed OFC's functional involvement in all schedules, but the effect size of inhibition is clearly largest in schedule 3 (the most uncertain schedule). Additionally, the largest effect size of M2 inhibition was in the

last schedule 1, the most certain schedule. We add these details to the abstract, Results, and Discussion.

Reviewer #1 (Remarks on code availability):

Unfortunately I did not have time to review the code.

Reviewer #2 (Remarks to the Author):

We appreciate the authors' efforts in conducting an additional experiment to break the inherent correlation between uncertainty and experience present in the original experimental design and address the causal role of OFC in decision making under conditions of high uncertainty. We agree that the new experimental design effectively disentangles uncertainty from experience. However, we find that the evidence from presented results could be stronger with improved statistical analysis.

We thank the reviewer for their feedback.

From our understanding, the authors calculated performance within a 15-trial moving window and treated each of these windows as independent samples. Using a moving average is concerning, as it creates repeated usage of the same data points. Each individual trial contributes to multiple (in this case, up to 15) overlapping windows. This can artificially inflate the sample size and generate high correlation across samples, thereby violating statistical assumptions of independence and potentially biasing statistical results.

We agree, and this was in error. Our previous work used non-overlapping windows for all accuracy statistics (Harris, Aguirre et al., 2021; Aguirre, Woo et al. 2024). We now conduct statistics on non-overlapping windows and find largely the same pattern of results (new Figure 5).

We suggest that using non-overlapping window would be more statistically appropriate. We appreciate that the GLM used here accounts for mixed effects and that is great. Nevertheless, given the complexity of the nested experimental design (windows nested within blocks, sessions, schedules, and rats), applying a hierarchical bootstrap method (<https://pmc.ncbi.nlm.nih.gov/articles/PMC7906290/>) could provide a more rigorous statistical test of significance. Alternatively, a more conservative approach would be to calculate the performance for each animal within each session and directly test whether the drug impacts this measure (each animal will be treated as a different independent sample in this case).

We thank the reviewer for this suggestion. We have used this more conservative, rigorous approach and replaced the statistics in this revision. Results yield, we think, cleaner and more convincing evidence of the differential involvement of these regions in learning under uncertainty.

Reviewer #2 (Remarks on code availability):

The code looks okay to me.

Reviewer #3 (Remarks to the Author):

We thank this Reviewer for their time and effort co-reviewing our work.

Reviewer #4 (Remarks to the Author):

The authors have carefully considered my concerns and provided detailed and satisfying responses to most of the comments I raised. In particular, they provided new data with decent numbers of animals, this new experiment not only provides a causal test of the distinct contributions of M2 and OFC in this task, also provides a well controlled paradigm to dissociate uncertainty across different schedules and learning.

We thank the reviewer for their feedback.

However, my 3rd major comment has not been fully addressed. In this comment, I raised concerns about the use of separate regression models may not be the best way to identify neurons as selective for “chosen side,” “trial outcome,” and “reward retrieval.” While I appreciate the authors’ effort to mitigate this issue by selecting narrow time windows, and also conducted an additional regression analysis, but these approaches do not entirely resolve the concern. From Figure 3A, it appears that there’s still lots of choice-related activities does not return to around 0 within the time window selected by the authors (+1s relative to choice). This raises ambiguity about whether the narrow time windows the author selected for calculating single cell selectivity is still overlap across these different variables, particularly in the block design task. If choice-related information persists, it may be mistakenly attributed to outcome or reward selectivity in this case. To more clearly dissociate the selectivity, fit both “chosen side” and “trial outcome” within a single regression model, or at least consider including “chosen side” as random effect when calculating selectivity for “trial outcome”/“reward retrieval”. Or the authors could show that almost all choice-selective neurons show no choice selectivity during the “trial outcome” or “reward retrieval” time windows.

We thank the reviewer for this suggestion. We regret not having fully addressed the previous comments. We have now re-run the statistical analyses to include choice and outcome within the same GLM formula. Results of single-cell selectivity are actually cleaner than before: i.e., M2 is unchanging, and only OFC neurons increase in selectivity across uncertainty schedules, now on all 3 measures- not just 2.

REVIEWER COMMENTS

Reviewer #1:

I appreciate the authors responding to my additional concerns. However, I am not convinced that what they are observing in the chemogenetic experiment is not just an order effect. This needs to be addressed or at the very least acknowledged in the manuscript as it is a significant limitation in the interpretation of the results.

We thank this reviewer for advising us to use caution in our interpretation and claims. We now include the following in the Discussion:

A limitation in the present experimental design is that we inactivated OFC or M2 neurons on the first experience with each schedule (i.e., CNO was administered on the first and VEH was administered on the second of two sessions within each schedule). While there is converging evidence that OFC is most involved in first reversals^{24, 31, 45} we were mainly interested in the pattern across schedules, and across M2 vs. OFC, not within a schedule. Indeed, we found the pattern of learning attenuations was different across schedules, and the two regions supported different facets of learning.

Reviewer #2:

Despite the authors' revised analysis, our original concern remains unresolved. In the previous revision, authors used a 15-trial sliding window to calculate performance $P(\text{better})$. This method reuses each trial multiple times, leading to non-independent samples. Because these performance estimates were then used in statistical tests, this repeated use of data violates the assumption of sample independence in statistical tests.

In the rebuttal, the authors claimed to have addressed this issue by using a non-overlapping window. However, according to the methods section, the revised method appears to compute $P(\text{better})$ at each trial by averaging all choices from the first trial up to the current one. This essentially means that the first trial contributes to the estimates for trials 1 through n . This is not a non-overlapping window. It is a maximally overlapping one. Therefore, the revised method still results in non-independent samples, and the statistical concern remains unresolved.

We sincerely thank the reviewer for pointing out this error, as we indeed made the problem of non-independent samples worse by using that cumulative averaging matlab function for the first time. Now in this revision wherever we report $p(\text{corr})$ trial-by-trial accuracy (Figure 1/Table 1 and Figure 5/Tables 18-21), we now use the raw data (0,1) for the trial-by-trial choices and analyze these data using a GLM with binomial distribution. The results are statistically identical to our R1 results with sliding windows (prior to our use of the cumulative average) showing that OFC neurons are causally involved in supporting learning across all schedules, whereas M2 neurons are only involved in Schedule 1.

Reviewer #3:

We thank this reviewer for co-reviewing our manuscript.

Reviewer #4:

The authors have carefully addressed all of my concerns, and the main results are still good enough to support the different roles of M2 and OFC under different uncertainty conditions in their task. I support publication of this study.

We thank this reviewer for co-reviewing our manuscript.

Minor comment:

Typo line 317 "We als find..." should be "We also find"

We have fixed this typo.

REVIEWERS' COMMENTS

Reviewer #2

I appreciate the authors addressing the concerns I raised. The authors now fit a glm to predict choice (1=better option, 0=worse option), and then use that predicted output from glm (pbetter) for their significance test of modulation effect. So instead of using the raw choice that animals made to test significance of modulation, they use inferred pbetter for the significance test. This is fine but the authors should consider mentioning this important analytical choice and its effect on the interpretation of their results.

We used mixed-effects general linear models (GLMs) to more rigorously handle data interdependencies (de Melo et al, 2022; Yu et al., 2022). This inference the reviewer refers to is a feature of GLM. We do feel this is the most rigorous statistical practice and respectfully disagree that we should add additional details of its limitations. We have various published reports using GLM to analyze pcorr (or pbetter):

<https://pubmed.ncbi.nlm.nih.gov/37456140/>

<https://pubmed.ncbi.nlm.nih.gov/34460275/>

<https://pubmed.ncbi.nlm.nih.gov/32555668/>

<https://pubmed.ncbi.nlm.nih.gov/31624264/>

<https://pubmed.ncbi.nlm.nih.gov/37968116/>

Reviewer #3

I co-reviewed this manuscript with one of the reviewers who provided the listed reports. This is part of the NatureCommunications initiative to facilitate training in peer review and to provide appropriate recognition for Early Career Researchers who co-review manuscripts.

We thank this reviewer for co-reviewing our manuscript.